# In-Place Feedback: A New Paradigm for Guiding LLMs in Multi-Turn Reasoning

## Abstract

Large language models (LLMs) are increasingly studied in the context of multi-turn reasoning, where models iteratively refine their outputs based on user-provided feedback. Such settings are crucial for tasks that require complex reasoning, yet existing feedback paradigms often rely on issuing new messages. LLMs struggle to integrate these reliably, leading to inconsistent improvements. In this work, we introduce *in-place feedback*, a novel interaction paradigm in which users directly edit an LLM's previous response, and the model conditions on this modified response to generate its revision. Empirical evaluations on diverse reasoning-intensive benchmarks reveal that in-place feedback achieves better performance than conventional multi-turn feedback while using 79.1% fewer tokens. Complementary analyses on controlled environments further demonstrate that in-place feedback resolves a core limitation of multi-turn feedback: models often fail to apply feedback precisely to erroneous parts of the response, leaving errors uncorrected and sometimes introducing new mistakes into previously correct content. These findings suggest that in-place feedback offers a more natural and effective mechanism for guiding LLMs in reasoning-intensive tasks.

## 1 Introduction

Large language models (LLMs) are increasingly positioned as in multi-turn conversations, where their effectiveness is measured by how well they generate responses that align with user intentions (Lee et al., 2022; Wang et al., 2025; Nath et al., 2025; Zhou et al., 2025; Kim et al., 2025). An example from chess demonstrates that a weaker yet cooperative agent can enable a player to outperform an opponent who is paired with a stronger but uncooperative agent (Hamade et al., 2024). Such findings highlight the growing importance of effectively incorporating user guidance in collaborative LLMs (Wu et al., 2025; Maheshwary et al., 2025).

Building on this perspective, we investigate a core mechanism of collaboration, feedback. In particular, we study how feedback can be used for error correction in multi-turn reasoning. Users can provide turn-level feedback through corrections, additional constraints, or supplemental information. For example, in a mathematical reasoning task, a user may identify an error in an LLM response and provide feedback to correct it. Unfortunately, recent studies have shown that LLMs often fail to incorporate user feedback in multi-turn interactions (Laban et al., 2025; Jiang et al., 2025; Sirdeshmukh et al., 2025). Expanding on these studies, we observe three failure modes when LLMs attempt to incorporate feedback: 1) previously correct content becomes incorrect after feedback, 2) the model disregards the feedback and repeats its earlier output, and 3) feedback is applied, but the subsequent reasoning steps introduce new errors.

To address these feedback integration failures, we propose *in-place feedback*, a novel interaction paradigm that reframes feedback not as a new instruction, but as a direct state repair. In this approach, user feedback is applied as an edit to the previous output, and the task of LLMs is to continue the generation from this corrected state. This method not only preserves correct prior reasoning but also constrains the model to build upon user-validated information. Through empirical evaluation on GPQA (Rein et al., 2024; Chandak et al., 2025), MMLU-pro (Wang et al., 2024; Chandak et al., 2025), and MATH-hard (Hendrycks et al., 2021) benchmarks, we show that in-place feedback achieves substantially improved performance and greater token efficiency compared to the standard multi-turn feedback paradigm.

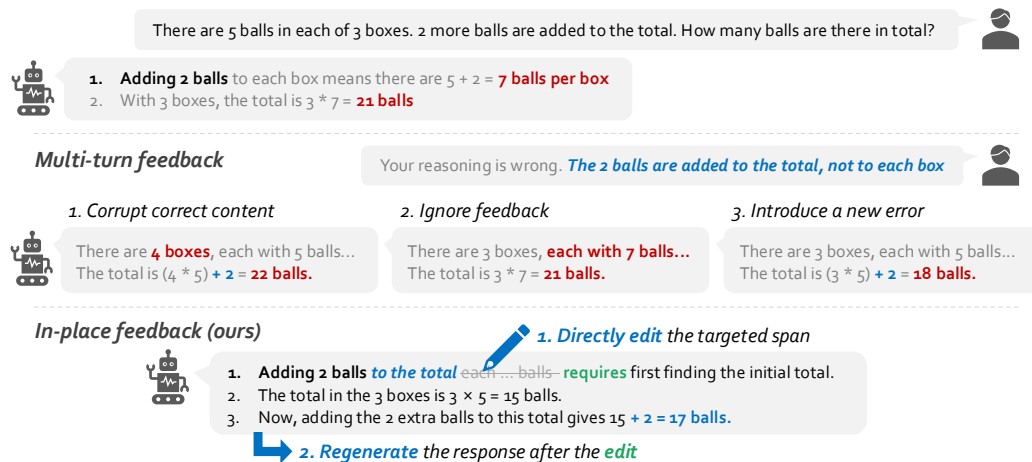

Figure 1: Illustration of common failure cases in multi-turn refinement and in-place feedback. After in-place feedback, the LLM continues generation from the green word "requires".

We also conduct a fine-grained analysis of the feedback paradigms on ZebraLogic (Lin et al., 2025), where reasoning errors and feedback can be extracted in a rule-based manner, to identify the precise factors underlying the effectiveness of in-place feedback. We observe that in multi-turn interaction, LLMs become less effective at incorporating feedback as the number of turns increases. In contrast, in-place feedback integrates corrections more effectively than multi-turn feedback, particularly in later turns. Moreover, in-place feedback propagates improvements through later reasoning steps, surpassing multi-turn feedback in overall refinement.

## 2 IN-PLACE FEEDBACK

### 2.1 MULTI-TURN REFINEMENT WITH FEEDBACK

We describe how feedback from humans or automated agents is incorporated into LLMs in interactive settings, focusing on multi-turn interactions. Let $\mathcal{M}$ be a target LLM. Given a problem $x$, the LLM produces an initial response as $y_0 = \mathcal{M}(x)$. Based on the problem $x$ and the initial response $y_0$, feedback is then generated to address potential reasoning errors in the initial response. Such feedback can be formalized by a function $\mathcal{F}$, yielding $f_0 = \mathcal{F}(x, y_0)$. In the subsequent turn, the target LLM refines its initial response using the feedback and generates the next response conditioned on the problem, the initial response, and the feedback, as $y_1 = \mathcal{M}(x, y_0, f_0)$. This illustrates the refinement process, in which each response is conditioned not only on the problem but also on the preceding response and its associated feedback.

More generally, the refinement extends to a multi-turn setting, where the LLM iteratively produces responses and incorporates feedback across multiple cycles: $y_t = \mathcal{M}(x, y_0, f_0, y_1, f_1, \ldots, y_{t-1}, f_{t-1})$, where $f_i = \mathcal{F}(x, y_i)$ denotes the feedback associated with the $i$-th response. We refer to this process as refinement with standard multi-turn feedback, which we hereafter simply call *multi-turn feedback*.

### 2.2 MOTIVATION: FAILURE CASE OF MULTI-TURN REFINEMENT

Recent work shows that LLMs often fail to reliably integrate user feedback (Laban et al., 2025; Jiang et al., 2025). Figure 1 illustrates common failure cases of multi-turn refinement. We observe three recurring failure modes: 1) previously correct content becomes incorrect after the feedback, 2) the model ignores the feedback and repeats its previous output, and 3) the feedback is applied but causes errors in subsequent reasoning steps. We hypothesize that these failures stem from regenerating the entire response from scratch at each turn. This process may overwrite correct reasoning and weaken the alignment between the feedback and the reasoning context it is meant to correct. We provide some examples of the failure cases of multi-turn refinement in Appendix D.2.

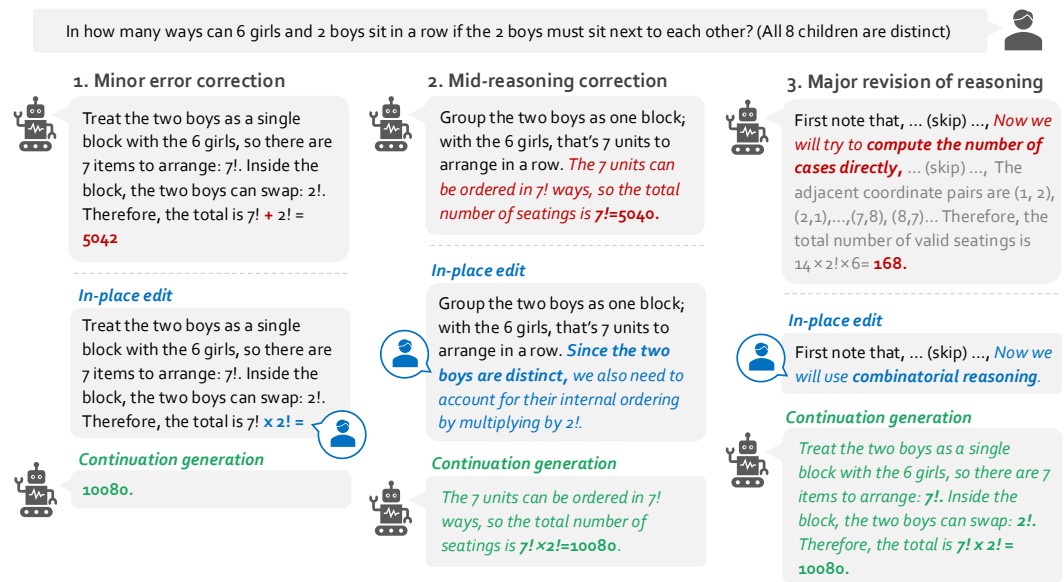

Figure 2: Representative examples of in-place feedback on a toy problem. Red marks incorrect reasoning, blue indicates the user corrections with in-place feedback, and green shows the subsequent reasoning based on the corrected context. Additional examples are provided in Appendix D.1.

This hypothesis highlights three requirements for effective refinement. Edits from user feedback should 1) remain focused on the targeted reasoning step, 2) preserve previously correct content outside this span, and 3) guide future reasoning from the corrected state rather than an outdated one. These considerations naturally lead us to ask: *Can we mitigate the above failures by letting the user directly edit the targeted span and constraining the model to continue generation from that point?*

## 2.3 IN-PLACE FEEDBACK

To address this question, we propose *in-place feedback*, a new multi-turn interaction mode that treats feedback as a state repair rather than a new instruction. As illustrated in Figure 1, our method proceeds in two stages. The first, *in-place edit*, allows the user to directly modify the model's previous response. The user then prunes the reasoning context that depends on the corrected span, while leaving the rest unchanged. In our setting, we assume the user identifies one or two mistakes in the reasoning and corrects only those parts. The second, *continuation generation*, regenerates only what is necessary to continue from the updated context. Together, these stages limit unintended changes and rebuild reasoning from the correction.

To illustrate how this method works in practice, Figure 2 presents representative cases of in-place feedback. For example, in math problems, in-place feedback can fix simple arithmetic mistakes or adjust flawed intermediate steps. In more complex cases, it can realign an incorrect reasoning path by revising larger portions of the solution.

**Benefits of in-place feedback.** Standard multi-turn feedback appends new turns to the history, causing early mistakes to persist and propagate across later reasoning. In-place feedback instead applies edits directly to the current output. By anchoring unchanged spans and updating only the edited portion, in-place feedback prevents error propagation. It also maintains global coherence and preserves a clear trace from the user's edit to the subsequent reasoning of LLMs.

In-place feedback also benefits from token efficiency. Standard multi-turn feedback accumulates a lengthy dialogue history and leads the model to regenerate entire reasoning chains, including parts that are already correct. In contrast, in-place feedback keeps the history compact by editing only the targeted span and continues generation from a corrected span, avoiding unnecessary regeneration. As a result, it reduces both input and output tokens, even under repeated feedback.

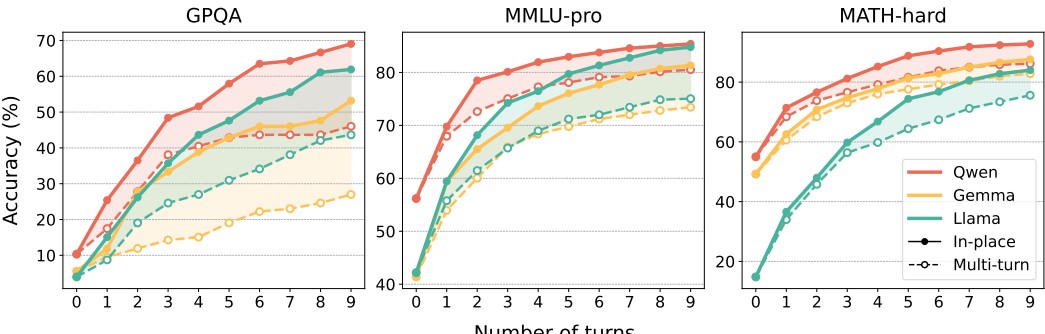

Figure 3: Comparison of in-place and multi-turn accuracies across models in MATH-hard, MMLU-pro, and GPQA. Across all datasets and LLM models, our in-place feedback approach consistently outperforms the multi-turn based feedback approach.

## 3 EMPIRICAL STUDY ON THE EFFECT OF IN-PLACE FEEDBACK

In this section, we evaluate the effectiveness of in-place feedback in multi-turn reasoning. We compare it with standard multi-turn feedback across multiple real-world datasets and LLMs.

### 3.1 EXPERIMENTAL SETUP

**Datasets and evaluation.** We conduct experiments on MATH-hard (Hendrycks et al., 2021), MMLU-pro free-form (Wang et al., 2024; Chandak et al., 2025), and GPQA free-form (Rein et al., 2024; Chandak et al., 2025). For MATH-hard, we sample 500 level-5 problems from MATH (Hendrycks et al., 2021). For MMLU-pro and GPQA, we use the free-form subsets introduced by Chandak et al. (2025), which contain only open-ended questions. We evaluate model answers in two stages. We first attempt exact matching. If it fails, we then apply an LLM judge to identify semantically equivalent answers expressed in different forms. We use GPT-oss-20b (OpenAI Team, 2025) as the judge model. Details on datasets and judge prompts are provided in Appendix A.

**Feedback function and agent for in-place feedback.** For experimental evaluation, it is necessary to automate the process of generating and applying feedback, which would otherwise require human intervention. Given a problem, its ground-truth solution, and the reasoning process of the LLM, the feedback function identifies the earliest critical error and generates a correction. The feedback function is designed to operate in both multi-turn and in-place settings.

For in-place feedback, a human should apply feedback directly to the previous response of the LLM. To automate this process in our experiments, we utilize an in-place feedback agent. The agent takes the feedback and the response of the LLM, identifies the sentence to be replaced, and provides its replacement. We then substitute the sentence and remove all subsequent text, since it may depend on the corrected span. We use GPT-5-mini (OpenAI, 2025) for both the feedback function and the in-place feedback agent. Further details on post-processing steps and prompt templates are provided in Appendix A.

**LLMs and hyperparameters.** We use three open-source LLMs: `Gemma-3-4b-it` (Gemma Team, 2025), `Qwen2.5-7B-Instruct` (Qwen Team, 2025), and `Llama-3.1-8B-Instruct` (Kassianik et al., 2025). Each model is evaluated for 10 turns with temperature set to 0. Further experimental settings are provided in Appendix A.

### 3.2 RESULTS

**Task performance.** Figure 3 shows how accuracy changes under multi-turn and in-place feedback as the number of turns increases. Across all datasets and models, in-place feedback consistently achieves higher accuracy and exhibits faster improvement over turns. On GPQA with `Gemma`, for

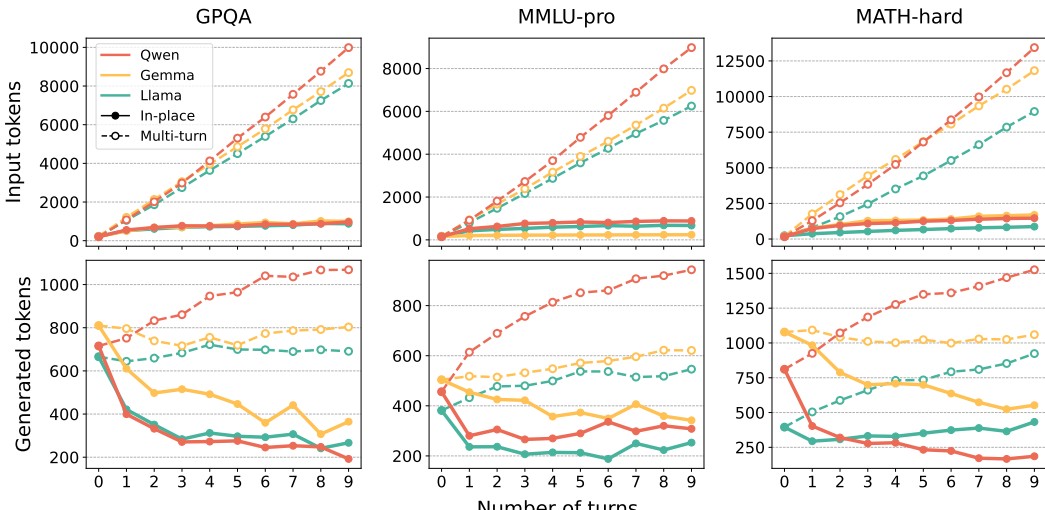

Figure 4: Number of input and generated tokens across multiple turns. In-place feedback consistently requires fewer tokens than multi-turn feedback across all datasets and LLMs.

example, in-place feedback achieves 53% accuracy, almost twice the performance of multi-turn feedback. On MMLU-pro, its accuracy at turn 5 already surpasses the final-turn performance of multi-turn feedback across all models. These results demonstrate that in-place feedback provides a more effective way to integrate external corrections into the reasoning of LLMs. We provide the qualitative examples in Appendix B.

**Token efficiency.** Beyond task performance, in-place feedback also exhibits superior efficiency in both input and generated token usage. Figure 4 presents the number of input and generated tokens across datasets. The result shows that in-place feedback requires substantially fewer tokens than multi-turn feedback. For input tokens, multi-turn feedback appends new turns to the dialogue history, causing token usage to grow linearly with the number of turns. In contrast, since in-place feedback does not accumulate the full dialogue history, the number of input tokens remains at a stable level. For generated tokens, in-place feedback preserves correct reasoning and revises only the erroneous parts, whereas multi-turn feedback generates entire reasoning steps from scratch. As a result, in-place feedback consistently produces shorter generations across turns. Aggregating input and output tokens, in-place feedback reduces token usage by 79.1% relative to multi-turn feedback, demonstrating substantially higher efficiency.

## 4 FEEDBACK EFFECTIVENESS IN CONTROLLED EXPERIMENTS

Prior work has mainly evaluated feedback in multi-turn interactions by measuring whether the final answer improves after feedback (Jiang et al., 2025; Sirdeshmukh et al., 2025). Such task-level evaluation leaves open *how feedback actually influences the reasoning process across turns*. Without analyzing turn-level dynamics, it is unclear whether models are using feedback or simply regenerating new responses. To address this gap, we design controlled experiments with ZebraLogic (Lin et al., 2025), where feedback is generated through a rule-based manner. Using this setting, we compare how multi-turn and in-place feedback incorporate corrections over successive turns and highlight where in-place feedback provides advantages.

### 4.1 SETUP FOR CONTROLLED EXPERIMENTS

**Task.** We conduct experiments on the ZebraLogic (Lin et al., 2025), a collection of 573 logic grid puzzles designed to evaluate the reasoning capability of LLMs. Each puzzle consists of $N$ houses and $M$ attributes such as *Name*, *Drink*, and *Hobby*, forming an $N \times M$ grid of cells. Attributes must take $N$ distinct values under uniqueness constraints, resulting in each cell having a single correct

value. A set of natural-language clues specifies additional logical relations, and the task is to assign values to all cells so that all constraints are satisfied. Details of the dataset are in Appendix A.

**Feedback functions.** We construct two rule-based feedback functions that differ in the amount of corrective information they provide: *Oracle* and *Top-k*. 1) *Oracle* reveals every incorrectly predicted cell along with its correct value. 2) *Top-k* selects the $k$ cells that most strongly violate logical constraints, identified using a Z3 solver (De Moura & Bjørner, 2008), and provides their correct values. For example, if the model predicts `Name of house 2 = Alice` while the ground truth is `Eric`, the feedback specifies `Name of house 2 is Eric, not Alice`.

**In-place feedback agent.** The in-place feedback agent simulates a human editor by directly modifying the LLM's response during evaluation. In math problems, each reasoning step depends on the previous one, so a correction usually requires discarding subsequent steps. Zebra puzzles, however, follow a different structure. They involve parallel reasoning, in which multiple constraints must be satisfied simultaneously. As a result, later reasoning steps can remain valid even if earlier ones are incorrect. To handle this, we retain subsequent reasoning steps after in-place feedback is applied.

We first segment the model's response into the reasoning steps. The agent checks each step against the received feedback and edits the response when a directly mispredicted attribute value is specified (*e.g.*, changing `Name of House 2 = Alice` to `Name of House 2 = Eric`). If there are reasoning steps that depend on the mispredicted value identified by the feedback (*e.g.*, reasoning built on `Alice in House 2`), those dependent steps are removed to prevent error propagation. After applying these edits or deletions, the final solution is removed, and the prompt `Further reasoning:` is appended to encourage continuation of the reasoning process. We employ GPT-5-mini as the in-place feedback agent, following the rule prompt in Figure A8.

**LLMs and hyperparameters.** We use three open-source LLMs, consistent with the previous experiments. We set $k = 2$ and $k = 4$ for Top-$k$ feedback. All experiments are run with three seeds. Detailed experimental settings are provided in Appendix A.

**Metrics.** We evaluate performance using two classes of metrics. To measure overall task performance, we use grid-level and cell-level accuracy. To conduct a more fine-grained analysis of the multi-turn refinement process, we introduce three complementary metrics that measure correctness preservation, feedback incorporation, and reasoning-driven self-correction.

- **Grid-level accuracy**. The proportion of grid puzzles that are solved perfectly, *i.e.*, all cells match the solution.

- **Cell-level accuracy**. The average proportion of correctly predicted cells over all puzzles.

- **Correctness-Preserving Ratio (CPR).** The proportion of cells that are correct in $y_t$ and remain correct in $y_{t+1}$, relative to the total number of cells that are correct in $y_t$. *This metric evaluates whether the model can retain valid reasoning while applying updates.*

- **Feedback Acceptance Ratio (FAR).** The proportion of cells flagged by feedback $f_t$ that are corrected in $y_{t+1}$, relative to the total number of feedback-provided cells of $y_t$. *This metric captures the model's ability to incorporate explicit corrective signals.*

- **Correction Through Reasoning Ratio (CTRR).** The proportion of cells that are incorrect in $y_t$ but corrected in $y_{t+1}$, relative to the total number of incorrect cells in $y_t$ that are not indicated by feedback $f_t$. *This metric measures the extent to which the model can generalize beyond explicit feedback and improve its reasoning autonomously.*

## 4.2 Analysis of Feedback Utilization

**Task performance.** Figure 5 shows grid accuracy and cell accuracy as the number of feedback turns increases. In-place feedback consistently outperforms multi-turn feedback, as observed in previous experiments. Interestingly, the gap in cell accuracy is smaller than the gap in grid accuracy. This suggests that while multi-turn feedback encounters difficulties in correcting the remaining few cells during iterative refinements, in-place feedback is more effective in addressing these corrections.

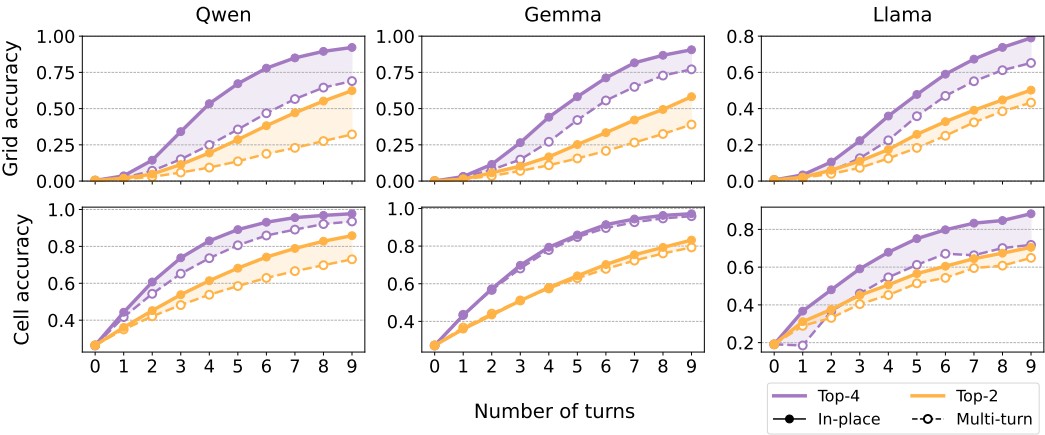

Figure 5: Grid and cell accuracy of LLMs on the Zebralogic dataset. Across both top-2 and top-4 feedback settings, in-place feedback consistently outperforms multi-turn feedback.

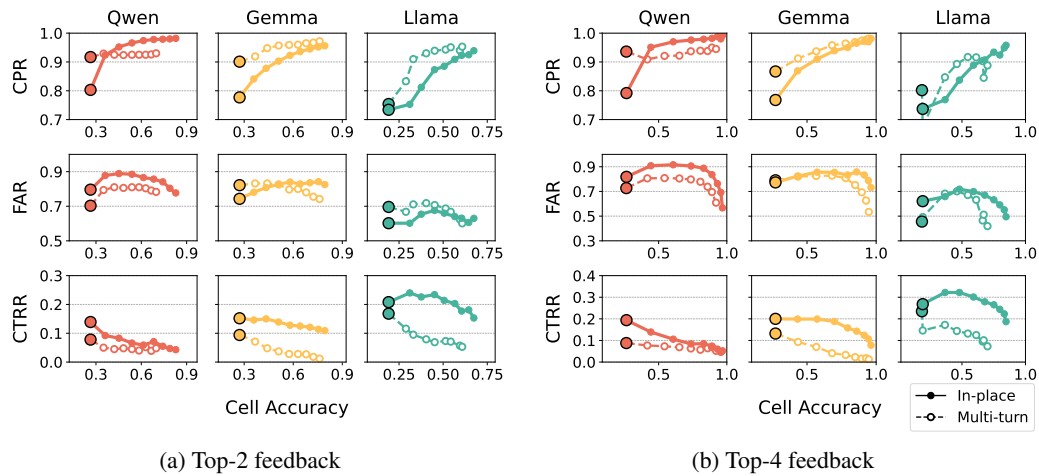

(a) Top-2 feedback

(b) Top-4 feedback

Figure 6: Correctness-Preserving Rate (CPR), Feedback Acceptance Rate (FAR), and Correction Through Reasoning Ratio (CTRR) for 10-turn conversations of LLMs on the ZebraLogic. The points with a black border represent the second response of the LLMs (i.e., $y_1$), and the subsequent responses across turns are connected by lines.

To gain a deeper understanding of this phenomenon, we examine the reasoning dynamics at the turn-level, focusing on how top-$k$ feedback is incorporated and influences the correction process. We also provide the results with the Oracle feedback in Appendix B.

**Turn-level dynamics of LLM behavior with multi-turn feedback.** Before analyzing the effect of in-place feedback, we first examine how LLMs behave under standard multi-turn feedback settings. This analysis highlights the dynamics when models incorporate feedback across multiple turns. We observe two distinct phases. In the initial phase, models are generally effective at incorporating feedback. However, they also exhibit a systematic tendency to modify portions of the response that are already correct, thereby compromising previously valid reasoning steps. As the conversation progresses, the models show increasing resistance to change, which reduces the effectiveness of further feedback. This behavioral shift is illustrated in Figure 6, where we analyze CPR, FAR, and CTRR alongside cell accuracy.

The capacity to preserve correct answers improves over successive turns, as reflected in the rising CPR values. In contrast, the ability to incorporate feedback exhibits a steady decline between turns

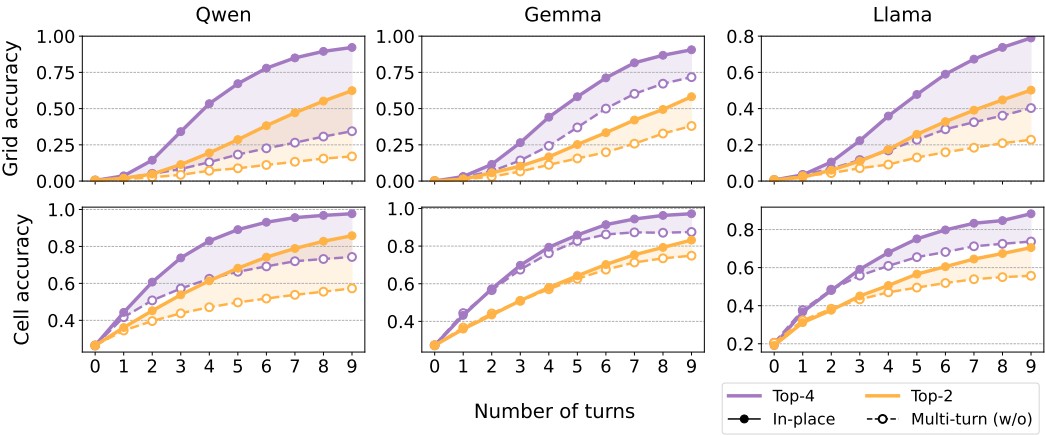

Figure 7: Grid and cell accuracy of LLMs on the ZebraLogic dataset without accumulated history compared against the in-place feedback. Multi-turn (w/o) shows the accuracy of responses only on the previous answer and the feedback, *i.e.*, $\tilde{y}_{t+1} = \mathcal{M}(x, y_t, f_t)$, without accumulated history.

5 and 9, as evidenced by the downward trend in FAR, suggesting that models become less receptive to feedback and more resistant to revising earlier responses. It is worth noting that the models are unlikely to generalize beyond the explicit feedback after the first few turns, as evidenced by lower CTRR. We also observe distinct behavior between different LLMs, despite showing similar overall trends. Specifically, Llama exhibits a comparatively higher CTRR, yet this comes at the cost of lower CPR and FAR relative to the other models.

**Advantage of in-place feedback.** We identify two key advantages of in-place feedback over multi-turn feedback. First, in-place feedback sustains the ability to incorporate feedback even when the number of turns increases, yielding higher FAR values than multi-turn feedback in later turns. This result accounts for the larger improvement observed in grid accuracy. Second, in-place feedback facilitates reasoning beyond the explicitly targeted errors, thereby leading to consistently higher CTRR. We conjecture that this improvement arises since in-place feedback reduces contextual interference from prior responses, allowing the model to more directly condition on the corrected span. By discarding subsequent content and regenerating from the point of modification, the model may better propagate the corrective signal to related parts of the reasoning process, thereby facilitating improvements even in cells not explicitly mentioned by the feedback.

The relatively lower CPR and FAR observed for in-place feedback during the earlier turns may reflect the effect of more intensive reasoning compared to multi-turn feedback. Moreover, due to the characteristics of the parallel reasoning problem, an in-place edited span may contain incorrect cell-related reasoning inherited from earlier turns under Top-$k$ feedback, which can propagate errors through subsequent reasoning. In contrast, under oracle feedback, where in-place edits could be error-free, both CPR and FAR are typically higher than in multi-turn feedback, except for CPR with Qwen in the first turn (see Figure A2).

### 4.3 EFFECT OF DIALOGUE HISTORY

As the dialogue progresses with multi-turn feedback, two factors may contribute to the observed decrease in FAR: 1) the accumulation of dialogue history that the model must condition on in a multi-turn approach, and 2) the reduction in the number of remaining incorrect cells as refinement progresses. We disentangle these two effects to better understand their respective contributions.

**Influence of accumulated history on feedback incorporation.** To investigate the effect of accumulated history, we refine the response by pruning the accumulated previous history, *i.e.*, $\tilde{y}_{t+1} = \mathcal{M}(x, y_t, f_t)$. Figure 7 presents the accuracy of the in-place feedback approach and the history-pruned variant. The results show that even when the accumulated history is removed, the feedback is still not properly incorporated in the subsequent turn. It is often suggested that users

open a new chat when the response does not align with their intent, yet our findings demonstrate that this approach does not offer a sufficient remedy. These results demonstrate that directly editing the LLM's response constitutes a more reliable and effective means of incorporating user feedback.

**Effect of the number of remaining incorrect cells on feedback incorporation.** We examine the top-4 feedback results in Figure 8, which report FAR together with the number of incorrect cells. Analyzing feedback–response pairs across all turns, we observe that FAR shows no significant correlation with the number of incorrect cells. This indicates that LLMs sustain a comparable level of feedback incorporation regardless of whether few or many incorrect cells remain.

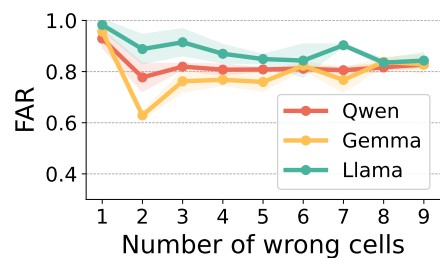

Figure 8: Change of FAR with respect to the number of incorrect cells. The x-axis denotes the number of incorrect cells in the previous LLM response across the entire puzzle, and FAR is measured under the setting where feedback for up to four cells is provided.

## 5 RELATED WORK

**Multi-turn interaction of LLMs.** Several studies aim to improve the performance of LLMs in multi-turn interaction. One line of work focuses on clarifying questions, where the LLM generates follow-up questions when the user's input is ambiguous (Zhang & Choi, 2025; Zamani et al., 2020; Aliannejadi et al., 2019). Zhang & Choi (2025) proposes a framework that integrates clarifying questions into the response generation process. Another line of work enhances multi-turn performance through training (Zhou et al., 2024; Shani et al., 2024; Wu et al., 2025). Wu et al. (2025), for example, fine-tunes LLMs with reinforcement learning to enhance their effectiveness. Our work focuses on how LLMs can achieve more effective interaction with users without additional training, while ensuring token efficiency.

**Refinement of LLMs.** Recent research explores self-refinement, an approach where the LLMs generate feedback on their own outputs and improve them accordingly. (Madaan et al., 2023; Dhuliawala et al., 2024; Shinn et al., 2023a; Nathani et al., 2023). Welleck et al. (2023) trains a separate model to produce feedback. Han et al. (2025) uses an external LLM agent to provide feedback for evaluating model performance. Zhang et al. (2025) employs a user simulation model to create interaction scenarios in multi-turn, which is closely related to our work. We show turn-level dynamics of LLMs in refinement with feedback and propose an alternative interaction scheme.

**Analysis on multi-turn conversations.** Recent studies analyze the performance of LLMs in multi-turn conversations. Jiang et al. (2025) shows that LLMs fail to reliably incorporate feedback, even when it is close to the correct answer. Laban et al. (2025) find that accuracy decreases when a single problem is divided into multiple parts and solved by LLMs in a multi-turn manner. Sirdeshmukh et al. (2025) analyzes the performance of LLMs in multi-turn conversations across four categories, including instruction retention and self-coherence. While these studies analyze the performance of LLMs in the multi-turn setting, they primarily report high-level metrics, such as overall task accuracy, without analyzing how reasoning evolves across turns or how errors propagate.

## 6 CONCLUSION

We introduce in-place feedback, an interaction method where users directly edit an LLM's prior response, and the model generates an output conditioned on this edited context. This approach achieves stronger refinement performance on reasoning benchmarks and is more efficient, requiring fewer input and output tokens. Through controlled experiments on ZebraLogic, we show that in-place feedback mitigates key challenges of multi-turn feedback. While our work focuses on reasoning tasks, we expect in-place feedback to be useful for a wide range of applications, such as document editing and code writing.

**Ethical consideration.** Our method allows users to edit an LLM response and then continue generating from that edit. However, such edits can be misused to bypass safety mechanisms. For example, a user might insert harmful instructions or unsafe text, similar to jailbreak attacks. A straightforward defense is to run user edits through a safety filter before resuming the generation process. We leave the design of defenses against such attacks to future work.

**Reproducibility statement.** We utilize four open-sourced LLMs and one closed-sourced LLM. All experimental settings and prompts are provided in Section 3.1, Section 4.1, and Appendix A to ensure reproducibility. For the closed-source LLM, we use the gpt-5-mini-2025-08-07 version.

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

# A    EXPERIMENTAL DETAILS

## A.1    DATASETS

### A.1.1    MMLU-PRO AND GPQA FREE-FORM

The MMLU-Pro and GPQA-Diamond datasets are employed to evaluate the knowledge-intensive reasoning abilities of LLMs in free-form question answering. Both datasets originate from multiple-choice benchmarks but are adapted for generative evaluation via answer matching.

**Motivation for free-form evaluation.**    Multiple-choice evaluation is efficient, but it has intrinsic limitations: models can exploit statistical patterns in the answer choices without engaging in critical reasoning. As a result, multiple-choice accuracy overestimates the model's ability to generate correct answers. By removing the choices and requiring free-form responses, models are forced to generate an answer directly, aligning the evaluation more closely with the capabilities that matter in real-world use cases.

**Selection of questions.**    MMLU-Pro is derived from the MMLU benchmark and initially includes approximately 12,000 questions across various domains. To ensure that the questions are answerable without the provided answer choices, an automatic filtering process is employed. This process uses a rubric-based grader to narrow down the dataset to about 5,500 questions. From this reduced pool, questions that are sufficiently specific and have a unique correct answer are selected. After this, 493 questions are manually filtered. This dataset offers comprehensive coverage across various domains, making it an ideal resource for evaluating general knowledge and reasoning skills.

The GPQA-Diamond dataset comprises 198 graduate-level science questions, designed to be challenging and test in-depth knowledge and critical reasoning. Similar to MMLU-Pro, a filtering process is applied to select questions with clear and specific correct answers, resulting in a final set of 126 questions. This dataset is more focused and rigorous, emphasizing high-quality scientific questions that demand deep reasoning.

These preparation steps yield two complementary free-form datasets: MMLU-Pro, which provides broader coverage across domains with 493 carefully filtered and annotated items, and GPQA-Diamond, which offers a smaller but more rigorous collection of 126 high-quality scientific questions. This ensures that free-form evaluations are conducted only on questions with clear, unambiguous solutions.

### A.1.2    ZEBRALOGIC

The ZebraLogic dataset comprises logic grid puzzles designed to assess the reasoning capabilities of LLMs. Each puzzle is structured around a grid with a certain number of houses and attributes. Specifically, each puzzle involves $N$ houses and $M$ attributes, creating an $N \times M$ grid that needs to be filled. The attributes in these puzzles are distinct for each house, where each attribute has $N$ unique values corresponding to the houses.

The puzzles come with a set of clues that impose logical constraints on the grid. For example, one clue might specify that "The person who likes milk is Eric", while another might state, "The person who drinks water is Arnold". These clues help guide the reasoning process to fill in the grid, ensuring that all constraints are satisfied. Importantly, each puzzle has a unique solution, which guarantees that any feedback provided for solving the puzzle is definitive and accurate. This structure ensures that ZebraLogic provides a rigorous framework for evaluating logical reasoning in models, where the set of clues provided uniquely determines each puzzle's solution.

**Puzzle generation.**    Puzzles are generated by first sampling a complete solution, then constructing a superset of consistent clues from a fixed inventory. The clue types are as follows: FOUNDAT, SAMEHOUSE, NOTAT, DIRECTLEFT/RIGHT, SIDEBYSIDE, LEFT/RIGHTOF, and ONE/TWOBETWEEN. Each clue type provides a constraint by capturing a specific relationship between variables. A minimal subset of clues is retained through iterative pruning while preserving

the uniqueness of the solution. This guarantees that puzzles are neither under-specified nor trivially over-constrained.

**Dataset filtering.** The ZebraLogic dataset contains 1,000 puzzles spanning all combinations of $N, M \in \{2, \ldots, 6\}$. We filter the dataset by puzzle difficulty based on the search space size, which is defined as the total number of possible configurations that satisfy only the uniqueness constraints of the puzzle; for a puzzle with $N$ houses and $M$ attributes, the search space size is $|\mathcal{S}| = (N!)^M$. Puzzles with small search spaces ($|\mathcal{S}| < 10^3$) are excluded, along with those of size $3 \times 4$ and $3 \times 5$, as well as invalid puzzles (*e.g.*, cases where distinct categories share identical attribute values). This yields a controlled, reasoning-intensive testbed for analyzing the feedback acceptance of LLMs in multi-turn settings.

**Input and output format.** Puzzles are presented in natural language, followed by an instruction asking the model to fill the $N \times M$ grid. The input template prompt is provided in Figure A6. The expected output is a structured JSON table. This format enables automatic cell-level evaluation and the application of fine-grained feedback. We attempt up to 30 re-generations to obtain a syntactically valid JSON output. If all attempts fail, we treat the instance as wrong and omit it from CPR, FAR, and CTRR calculations.

For JSON-parsed prediction, we employ a fuzz score from the Python rapidfuzz library to perform exact-matching evaluation. Specifically, we compute the highest similarity score among the candidate attributes, and if the score exceeds 50, we adopt the corresponding attribute as the predicted value.

**Feedback functions.** We generate rule-based feedback using a fixed template, as illustrated in Figure A7. For `Llama`, we additionally append reasoning guidance to the feedback, since the model frequently produces only the final JSON-formatted answer without including the corrected reasoning process.

### A.2 PROMPT AND POST-PROCESSING

**Empirical experiments.** We define answer leakage as any explicit revelation of the ground-truth answer within feedback or intervention outputs. We prevent answer leakage, following Jiang et al. (2025). To prevent leakage, we use the prompts in Figure A4 and Figure A5, and apply post-processing for each agents: (i) after generating feedback, we scan the message and mask any span that reveals the ground truth; (ii) for the in-place feedback agent, if any part of the message exposes the solution, we prune those spans before presenting the message to the model.

**ZebraLogic.** We constrain the in-place feedback agent to avoid introducing reasoning beyond the scope of the provided feedback. The model's output is segmented into discrete reasoning steps, which are then checked for consistency with the provided feedback. In cases of conflict, the corresponding step is minimally revised, following the prompt in Figure A8. Notably, the agent does not have direct access to the puzzle itself, which limits its ability to extend reasoning beyond the explicitly given feedback.

### A.3 HYPERPARAMETERS.

We observed that `Llama-3.1-8B-Instruct` can be overly verbose, resulting in degradation of generation quality. To stabilize decoding, we set its repetition penalty to 1.15 for `Llama`. All other models use a repetition penalty of 1.0. The maximum generation length is 2048 tokens for all models and experiments. We use vLLM (Kwon et al., 2023) for efficient inference.

## B ADDITIONAL EXPERIMENTAL RESULTS

Figure A1 presents grid and cell accuracy as a function of the number of turns. In-place feedback converges substantially faster than multi-turn feedback, indicating that it enables the model to incorporate feedback more efficiently. Figure A2 illustrates LLM behavior under oracle feedback.

In most cases, in-place feedback achieves higher CPR and FAR compared to multi-turn feedback, without CPR of the `Qwen` in the first turn.

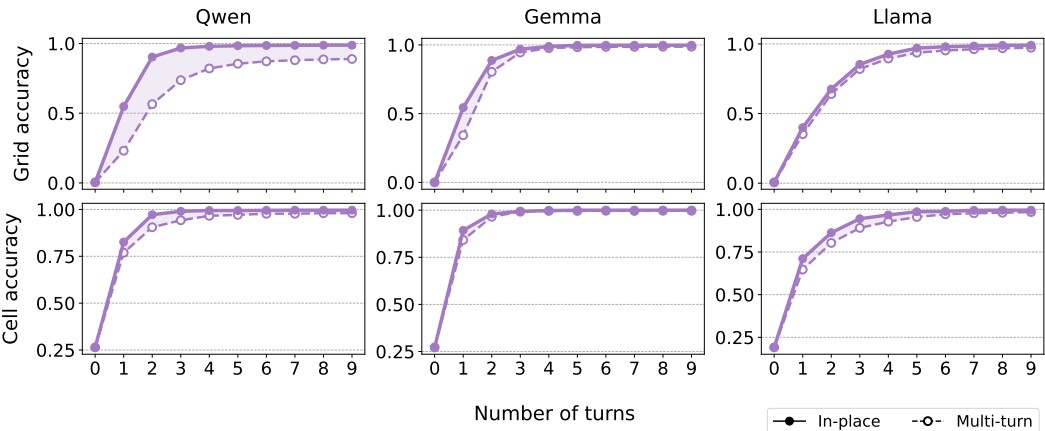

Figure A1: Grid and cell accuracy of LLMs on the Zebralogic dataset. In-place feedback consistently outperforms multi-turn feedback under an oracle setting.

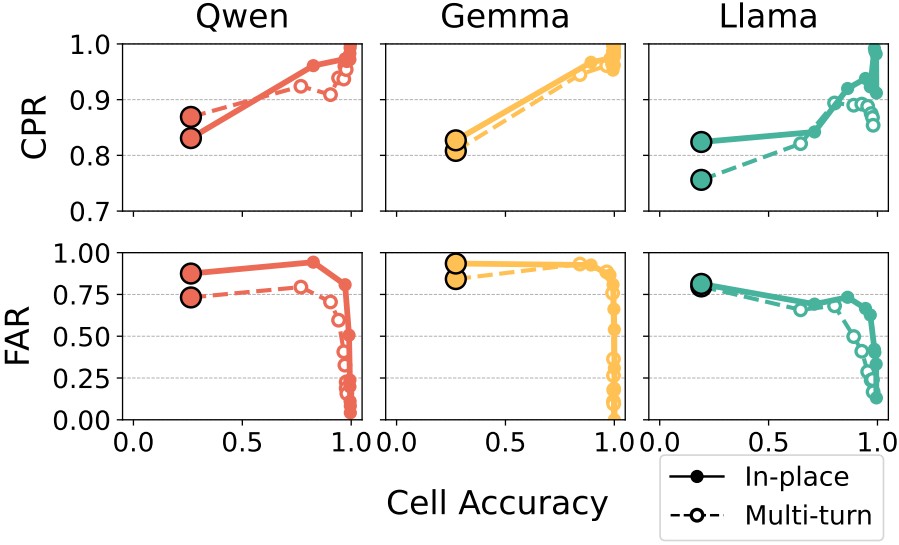

Figure A2: Correctness-Preserving Rate (CPR), Feedback Acceptance Rate (FAR), and Comparison of Correction Through Reasoning Ratio (CTRR) for 10-turn conversations of LLMs on the ZebraLogic. The Oracle feedback function is used. The points with a black border represent the second response of the LLMs (i.e., $y_1$), and the subsequent responses across turns are connected by lines.

## C    PERFORMANCE COMPARISION

### C.1    TASK PERFORMANCE

| Model | Method | Number of turns | | | | | | | | | |
|---|---|---|---|---|---|---|---|---|---|---|---|
| | | 0 | 1 | 2 | 3 | 4 | 5 | 6 | 7 | 8 | 9 |
| Qwen-2.5-7B | Multi-turn | 55.0 | 68.4 | 73.8 | 76.6 | 79.2 | 81.6 | 83.8 | 85.0 | 85.8 | 86.2 |
| | Reflexion | 55.0 | 62.6 | 66.2 | 69.2 | 71.2 | 72.8 | 74.2 | 75.0 | 76.4 | 77.4 |
| | In-place | 55.0 | 71.4 | 76.6 | 81.2 | 85.2 | 88.8 | 90.4 | 91.8 | 92.4 | 92.8 |
| Gemma-3-4B | Multi-turn | 49.2 | 60.6 | 68.4 | 73.0 | 76.0 | 77.6 | 79.2 | 80.4 | 82.0 | 82.8 |
| | Reflexion | 49.2 | 60.4 | 65.0 | 68.0 | 70.6 | 72.6 | 74.6 | 76.4 | 77.2 | 78.0 |
| | In-place | 49.2 | 62.6 | 70.8 | 74.8 | 77.8 | 81.4 | 82.8 | 85.0 | 86.6 | 87.6 |
| Llama-3.1-8B | Multi-turn | 14.8 | 34.0 | 45.8 | 56.4 | 59.8 | 64.4 | 67.4 | 71.2 | 73.4 | 75.6 |
| | Reflexion | 14.8 | 29.4 | 40.0 | 47.0 | 52.8 | 57.2 | 62.2 | 65.6 | 67.6 | 69.8 |
| | In-place | 14.8 | 36.6 | 48.0 | 59.8 | 66.8 | 74.4 | 76.8 | 80.6 | 82.8 | 84.0 |

Table A1: Performance comparison with Reflexion (Shinn et al., 2023b) on the Math-hard dataset.

| Model | Method | Number of turns | | | | | | | | | |
|---|---|---|---|---|---|---|---|---|---|---|---|
| | | 0 | 1 | 2 | 3 | 4 | 5 | 6 | 7 | 8 | 9 |
| Qwen-2.5-7B | Multi-turn | 56.2 | 68.0 | 72.6 | 75.1 | 77.3 | 78.1 | 79.1 | 79.3 | 80.1 | 80.5 |
| | Reflexion | 56.2 | 67.1 | 70.8 | 73.4 | 75.7 | 77.7 | 78.5 | 79.7 | 80.7 | 80.9 |
| | In-place | 56.2 | 69.8 | 78.5 | 80.1 | 81.9 | 83.0 | 83.8 | 84.6 | 85.0 | 85.4 |
| Gemma-3-4B | Multi-turn | 41.4 | 54.0 | 60.0 | 65.9 | 68.4 | 69.8 | 71.2 | 72.0 | 72.8 | 73.4 |
| | Reflexion | 41.4 | 52.9 | 57.2 | 63.5 | 66.3 | 68.0 | 69.6 | 70.4 | 71.6 | 72.2 |
| | In-place | 41.4 | 59.4 | 65.5 | 69.6 | 73.6 | 76.1 | 77.7 | 79.5 | 80.7 | 81.3 |
| Llama-3.1-8B | Multi-turn | 42.2 | 55.8 | 61.5 | 65.7 | 69.0 | 71.2 | 72.0 | 73.4 | 74.8 | 75.1 |
| | Reflexion | 42.2 | 57.0 | 63.9 | 68.6 | 71.2 | 74.0 | 75.9 | 77.3 | 78.3 | 78.7 |
| | In-place | 42.2 | 59.4 | 68.2 | 74.2 | 76.5 | 79.7 | 81.3 | 82.8 | 84.2 | 84.8 |

Table A2: Performance comparison with Reflexion (Shinn et al., 2023b) on the MMLU-Pro dataset.

| Model | Method | Number of turns | | | | | | | | | |
|---|---|---|---|---|---|---|---|---|---|---|---|
| | | 0 | 1 | 2 | 3 | 4 | 5 | 6 | 7 | 8 | 9 |
| Qwen-2.5-7B | Multi-turn | 10.3 | 17.5 | 27.8 | 38.1 | 40.5 | 42.9 | 43.7 | 43.7 | 43.7 | 46.0 |
| | Reflexion | 10.3 | 19.0 | 27.0 | 29.4 | 34.9 | 36.5 | 37.3 | 38.1 | 38.1 | 38.9 |
| | In-place | 10.3 | 25.4 | 36.5 | 48.4 | 51.6 | 57.9 | 63.5 | 64.3 | 66.7 | 69.0 |
| Gemma-3-4B | Multi-turn | 5.6 | 9.5 | 11.9 | 14.3 | 15.1 | 19.0 | 22.2 | 23.0 | 24.6 | 27.0 |
| | Reflexion | 5.6 | 8.7 | 10.3 | 13.5 | 15.9 | 17.5 | 18.3 | 22.2 | 22.2 | 23.8 |
| | In-place | 5.6 | 11.9 | 27.8 | 33.3 | 38.9 | 42.9 | 46.0 | 46.0 | 47.6 | 53.2 |
| Llama-3.1-8B | Multi-turn | 4.0 | 8.7 | 19.0 | 24.6 | 27.0 | 31.0 | 34.1 | 38.1 | 42.1 | 43.7 |
| | Reflexion | 4.0 | 8.7 | 15.1 | 20.6 | 23.0 | 27.8 | 31.7 | 34.1 | 37.3 | 40.5 |
| | In-place | 4.0 | 15.1 | 26.2 | 35.7 | 43.7 | 47.6 | 53.2 | 55.6 | 61.1 | 61.9 |

Table A3: Performance comparison with Reflexion (Shinn et al., 2023b) on the GPQA dataset.

## C.2 TASK PERFORMANCE OF LARGE-SCALE LLMs

| Model | Method | Number of turns | | | | |
|---|---|---|---|---|---|---|
| | | 0 | 1 | 2 | 3 | 4 |
| Llama-3.1-70B | Multi-turn | 31.2 | 52.6 | 63.2 | 71.0 | 74.6 |
| | Reflexion | 31.2 | 51.6 | 62.2 | 67.0 | 72.4 |
| | In-place | 31.2 | 53.6 | 64.6 | 71.8 | 76.8 |
| Gemma-3-27B | Multi-turn | 78.2 | 86.6 | 89.2 | 90.4 | 91.0 |
| | Reflexion | 78.2 | 86.2 | 87.8 | 89.8 | 90.8 |
| | In-place | 78.2 | 87.2 | 89.6 | 91.0 | 92.4 |

Table A4: Performance comparison with Reflexion (Shinn et al., 2023b) on the MATH-Hard dataset.

| Model | Method | Number of turns | | | | |
|---|---|---|---|---|---|---|
| | | 0 | 1 | 2 | 3 | 4 |
| Llama-3.1-70B | Multi-turn | 54.8 | 70.2 | 75.3 | 77.1 | 77.9 |
| | Reflexion | 54.8 | 68.8 | 76.5 | 78.5 | 80.2 |
| | In-place | 54.8 | 70.6 | 77.7 | 79.1 | 81.3 |
| Gemma-3-27B | Multi-turn | 61.3 | 73.4 | 76.3 | 79.3 | 80.5 |
| | Reflexion | 61.3 | 74.6 | 78.7 | 80.3 | 81.4 |
| | In-place | 61.3 | 74.4 | 79.3 | 81.7 | 83.4 |

Table A5: Performance comparison with Reflexion (Shinn et al., 2023b) on the MMLU-Pro dataset.

| Model | Method | Number of turns | | | | |
|---|---|---|---|---|---|---|
| | | 0 | 1 | 2 | 3 | 4 |
| Llama-3.1-70B | Multi-turn | 18.3 | 29.4 | 42.1 | 46.0 | 50.8 |
| | Reflexion | 18.3 | 30.2 | 40.5 | 46.6 | 50.8 |
| | In-place | 18.3 | 31.8 | 42.9 | 46.8 | 53.2 |
| Gemma-3-27B | Multi-turn | 17.5 | 34.1 | 42.9 | 48.4 | 51.6 |
| | Reflexion | 17.5 | 33.3 | 39.7 | 44.4 | 47.6 |
| | In-place | 17.5 | 34.9 | 43.7 | 49.2 | 58.7 |

Table A6: Performance comparison with Reflexion (Shinn et al., 2023b) on the GPQA dataset.

## C.3 OUTPUT TOKEN LENGTH

| Model | Method | Number of turns | | | | | | | | | |
|---|---|---|---|---|---|---|---|---|---|---|---|
| | | 0 | 1 | 2 | 3 | 4 | 5 | 6 | 7 | 8 | 9 |
| Qwen-2.5-7B | Multi-turn | 810.9 | 925.4 | 1072.1 | 1186.2 | 1276.7 | 1349.2 | 1360.0 | 1408.0 | 1469.4 | 1525.9 |
| | Reflexion | 810.9 | 850.7 | 945.5 | 987.7 | 1011.2 | 1020.3 | 1033.4 | 1031.4 | 1050.0 | 1071.5 |
| | In-place | 810.9 | 403.1 | 320.2 | 277.7 | 283.0 | 232.6 | 224.2 | 172.0 | 166.8 | 185.7 |
| Gemma-3-4B | Multi-turn | 1078.3 | 1092.7 | 1041.6 | 1011.4 | 1001.7 | 1023.5 | 998.8 | 1028.6 | 1025.2 | 1059.6 |
| | Reflexion | 1078.3 | 907.0 | 961.6 | 986.1 | 987.7 | 984.3 | 999.7 | 1041.3 | 1013.6 | 992.2 |
| | In-place | 1078.3 | 983.2 | 788.2 | 699.4 | 708.3 | 699.5 | 636.8 | 573.4 | 523.8 | 552.7 |
| Llama-3.1-8B | Multi-turn | 395.2 | 504.4 | 587.3 | 661.0 | 730.8 | 734.3 | 792.9 | 809.4 | 852.6 | 923.6 |
| | Reflexion | 395.2 | 479.1 | 588.1 | 647.8 | 696.7 | 705.3 | 743.1 | 790.1 | 844.8 | 861.5 |
| | In-place | 395.2 | 294.1 | 307.8 | 331.9 | 328.6 | 351.5 | 373.8 | 388.6 | 364.8 | 432.4 |

Table A7: Output token length on the Math-Hard dataset.

| Model | Method | Number of turns | | | | | | | | | |
|-------|--------|---|---|---|---|---|---|---|---|---|---|
| | | 0 | 1 | 2 | 3 | 4 | 5 | 6 | 7 | 8 | 9 |
| Qwen-2.5-7B | Multi-turn | 456.1 | 614.2 | 689.3 | 757.0 | 813.6 | 851.4 | 860.8 | 907.1 | 919.7 | 943.1 |
| | Reflexion | 456.1 | 597.1 | 678.3 | 721.4 | 736.5 | 780.6 | 766.5 | 794.5 | 847.6 | 833.7 |
| | In-place | 456.1 | 280.4 | 305.4 | 265.8 | 269.9 | 289.9 | 336.0 | 298.2 | 320.2 | 308.2 |
| Gemma-3-4B | Multi-turn | 504.2 | 518.1 | 514.7 | 531.9 | 548.0 | 571.1 | 578.8 | 596.0 | 622.1 | 621.1 |
| | Reflexion | 504.2 | 564.8 | 600.9 | 593.8 | 614.2 | 655.1 | 664.4 | 667.3 | 688.2 | 696.4 |
| | In-place | 504.2 | 454.4 | 425.5 | 421.8 | 357.2 | 373.2 | 349.3 | 406.4 | 358.9 | 342.1 |
| Llama-3.1-8B | Multi-turn | 381.4 | 432.4 | 477.4 | 480.0 | 499.4 | 537.3 | 537.0 | 514.7 | 518.2 | 546.3 |
| | Reflexion | 381.4 | 429.7 | 515.0 | 574.6 | 603.1 | 627.7 | 617.2 | 650.2 | 662.3 | 642.3 |
| | In-place | 381.4 | 236.2 | 236.7 | 206.9 | 214.0 | 212.8 | 188.0 | 249.9 | 223.5 | 253.1 |

Table A8: Output token length on the MMLU-Pro dataset.

| Model | Method | Number of turns | | | | | | | | | |
|-------|--------|---|---|---|---|---|---|---|---|---|---|
| | | 0 | 1 | 2 | 3 | 4 | 5 | 6 | 7 | 8 | 9 |
| Qwen-2.5-7B | Multi-turn | 715.5 | 751.6 | 832.8 | 860.5 | 946.8 | 964.7 | 1040.5 | 1035.5 | 1067.5 | 1068.6 |
| | Reflexion | 715.5 | 709.6 | 787.0 | 824.5 | 847.2 | 853.0 | 835.1 | 815.3 | 837.1 | 850.9 |
| | In-place | 715.5 | 399.5 | 332.2 | 270.9 | 272.5 | 275.7 | 245.1 | 253.2 | 247.8 | 192.1 |
| Gemma-3-4B | In-place | 810.3 | 610.4 | 497.5 | 515.1 | 491.7 | 446.4 | 359.6 | 441.3 | 306.9 | 364.7 |
| | Multi-turn | 810.3 | 795.9 | 739.2 | 716.3 | 755.4 | 718.7 | 773.2 | 786.6 | 791.7 | 803.5 |
| | Reflexion | 810.3 | 796.5 | 797.6 | 774.2 | 815.5 | 837.2 | 816.6 | 835.6 | 823.3 | 815.7 |
| Llama-3.1-8B | Multi-turn | 665.7 | 644.4 | 659.2 | 683.2 | 721.5 | 699.2 | 697.8 | 689.6 | 697.8 | 690.8 |
| | Reflexion | 665.7 | 614.8 | 671.1 | 678.5 | 667.1 | 673.4 | 660.3 | 646.5 | 651.6 | 627.4 |
| | In-place | 665.7 | 421.3 | 351.0 | 283.8 | 312.3 | 297.0 | 292.7 | 306.9 | 241.3 | 266.8 |

Table A9: Output token length on the GPQA dataset.

## C.4 OUTPUT TOKEN LENGTH OF LARGE-SCALE LLMS

| Model | Method | Number of turns | | | | |
|-------|--------|---|---|---|---|---|
| | | 0 | 1 | 2 | 3 | 4 |
| Llama-3.1-70B | Multi-turn | 279.8 | 380.0 | 482.5 | 522.4 | 593.4 |
| | Reflexion | 279.8 | 434.3 | 555.5 | 615.1 | 592.4 |
| | In-place | 279.8 | 249.2 | 239.0 | 315.2 | 300.5 |
| Gemma-3-27B | Multi-turn | 927.4 | 1134.5 | 1238.2 | 1154.2 | 1098.9 |
| | Reflexion | 927.4 | 969.4 | 1134.5 | 1222.9 | 1204.4 |
| | In-place | 927.4 | 807.4 | 731.0 | 879.4 | 756.5 |

Table A10: Output token length on the Math-Hard dataset.

| Model | Method | Number of turns | | | | |
|-------|--------|---|---|---|---|---|
| | | 0 | 1 | 2 | 3 | 4 |
| Llama-3.1-70B | Multi-turn | 310.1 | 385.4 | 423.6 | 467.4 | 477.6 |
| | Reflexion | 310.1 | 432.1 | 515.1 | 575.7 | 623.8 |
| | In-place | 310.1 | 153.9 | 160.8 | 166.7 | 147.7 |
| Gemma-3-27B | Multi-turn | 420.3 | 511.5 | 561.6 | 561.3 | 605.3 |
| | Reflexion | 420.3 | 586.8 | 680.9 | 712.1 | 723.7 |
| | In-place | 420.3 | 291.5 | 297.5 | 327.8 | 324.0 |

Table A11: Output token length on the MMLU-Pro dataset.

| Model | Method | Number of turns | | | | |
|-------|--------|------|------|------|------|------|
| | | 0 | 1 | 2 | 3 | 4 |
| Llama-3.1-70B | Multi-turn | 597.8 | 549.2 | 508.2 | 515.6 | 571.9 |
| | Reflexion | 597.8 | 574.9 | 607.8 | 600.5 | 620.3 |
| | In-place | 597.8 | 280.1 | 229.7 | 232.3 | 259.7 |
| Gemma-3-27B | Multi-turn | 738.9 | 641.1 | 643.1 | 666.4 | 679.0 |
| | Reflexion | 738.9 | 683.3 | 728.2 | 751.3 | 749.8 |
| | In-place | 738.9 | 417.1 | 330.4 | 292.9 | 315.6 |

Table A12: Output token length on the GPQA dataset.

## C.5 INPUT TOKEN LENGTH

| Model | Method | Number of turns | | | | | | | | | |
|-------|--------|------|------|------|------|------|------|------|------|------|------|
| | | 0 | 1 | 2 | 3 | 4 | 5 | 6 | 7 | 8 | 9 |
| Qwen-2.5-7B | Multi-turn | 159.6 | 1288.6 | 2538.6 | 3838.7 | 5235.0 | 6806.2 | 8368.0 | 9977.5 | 11671.0 | 13434.7 |
| | Reflexion | 159.6 | 1364.8 | 858.4 | 1134.3 | 1100.5 | 1144.9 | 1153.2 | 1160.6 | 1095.7 | 1125.5 |
| | In-place | 159.6 | 734.7 | 950.6 | 1088.9 | 1135.6 | 1250.6 | 1302.8 | 1392.1 | 1439.3 | 1464.7 |
| Gemma-3-4B | Multi-turn | 185.8 | 1764.0 | 3117.7 | 4440.7 | 5591.1 | 6843.3 | 8062.8 | 9342.5 | 10515.5 | 11818.4 |
| | Reflexion | 185.8 | 1898.6 | 1537.5 | 1712.3 | 1717.8 | 1737.9 | 1712.2 | 1744.5 | 1739.2 | 1720.0 |
| | In-place | 185.8 | 767.0 | 1019.0 | 1286.6 | 1306.8 | 1329.0 | 1401.7 | 1594.1 | 1639.5 | 1691.5 |
| Llama-3.1-8B | Multi-turn | 159.6 | 1288.6 | 2538.6 | 3838.7 | 5235.0 | 6806.2 | 8368.0 | 9977.5 | 11671.0 | 13434.7 |
| | Reflexion | 159.6 | 826.2 | 1073.4 | 1319.3 | 1361.2 | 1414.1 | 1413.2 | 1468.8 | 1515.9 | 1566.9 |
| | In-place | 159.6 | 734.7 | 950.6 | 1088.9 | 1135.6 | 1250.6 | 1302.8 | 1392.1 | 1439.3 | 1464.7 |

Table A13: Input token length on the Math-Hard dataset.

| Model | Method | Number of turns | | | | | | | | | |
|-------|--------|------|------|------|------|------|------|------|------|------|------|
| | | 0 | 1 | 2 | 3 | 4 | 5 | 6 | 7 | 8 | 9 |
| Qwen-2.5-7B | Multi-turn | 144.8 | 918.2 | 1809.5 | 2724.6 | 3698.0 | 4786.0 | 5804.9 | 6891.6 | 7984.2 | 8977.5 |
| | Reflexion | 144.8 | 947.4 | 1125.8 | 1282.0 | 1318.7 | 1333.7 | 1367.8 | 1345.4 | 1415.4 | 1449.1 |
| | In-place | 144.8 | 519.4 | 625.5 | 764.1 | 794.8 | 833.9 | 807.9 | 861.3 | 886.9 | 882.1 |
| Gemma-3-4B | Multi-turn | 165.1 | 931.1 | 1666.3 | 2384.0 | 3163.0 | 3895.5 | 4602.5 | 5360.0 | 6149.4 | 6980.5 |
| | Reflexion | 161.1 | 1005.2 | 1078.5 | 1191.2 | 1210.4 | 1217.3 | 1256.2 | 1272.8 | 1267.5 | 1285.5 |
| | In-place | 161.1 | 198.4 | 212.0 | 218.6 | 222.9 | 227.4 | 230.2 | 233.8 | 236.7 | 237.7 |
| Llama-3.1-8B | Multi-turn | 165.1 | 800.4 | 1473.2 | 2164.4 | 2871.0 | 3594.4 | 4274.1 | 4962.2 | 5578.6 | 6248.8 |
| | Reflexion | 165.1 | 829.6 | 973.4 | 1157.5 | 1209.8 | 1224.9 | 1256.0 | 1237.0 | 1249.4 | 1279.0 |
| | In-place | 165.1 | 422.1 | 487.9 | 534.0 | 592.6 | 625.2 | 665.1 | 638.6 | 677.9 | 670.8 |

Table A14: Input token length on the MMLU-Pro dataset.

| Model | Method | Number of turns | | | | | | | | | |
|-------|--------|------|------|------|------|------|------|------|------|------|------|
| | | 0 | 1 | 2 | 3 | 4 | 5 | 6 | 7 | 8 | 9 |
| Qwen-2.5-7B | Multi-turn | 203.6 | 1079.9 | 2013.6 | 2982.9 | 4127.7 | 5304.3 | 6395.3 | 7570.8 | 8763.9 | 9987.4 |
| | Reflexion | 203.6 | 1109.2 | 1258.6 | 1484.6 | 1521.4 | 1541.4 | 1544.0 | 1519.1 | 1511.7 | 1525.8 |
| | In-place | 203.6 | 553.7 | 678.3 | 771.6 | 759.5 | 766.1 | 847.4 | 848.5 | 876.1 | 966.4 |
| Gemma-3-4B | Multi-turn | 185.8 | 1764.0 | 3117.7 | 4440.7 | 5591.1 | 6843.3 | 8062.8 | 9342.5 | 10515.5 | 11818.4 |
| | Reflexion | 185.8 | 1898.6 | 1537.5 | 1712.3 | 1717.8 | 1737.9 | 1712.2 | 1744.5 | 1739.2 | 1720.0 |
| | In-place | 185.8 | 767.0 | 1019.0 | 1286.6 | 1306.8 | 1329.0 | 1401.7 | 1594.1 | 1639.5 | 1691.5 |
| Llama-3.1-8B | Multi-turn | 159.6 | 1288.6 | 2538.6 | 3838.7 | 5235.0 | 6806.2 | 8368.0 | 9977.5 | 11671.0 | 13434.7 |
| | Reflexion | 159.6 | 826.2 | 1073.4 | 1319.3 | 1361.2 | 1414.1 | 1413.2 | 1468.8 | 1515.9 | 1566.9 |
| | In-place | 159.6 | 734.7 | 950.6 | 1088.9 | 1135.6 | 1250.6 | 1302.8 | 1392.1 | 1439.3 | 1464.7 |

Table A15: Input token length on the GPQA dataset.

## C.6 INPUT TOKEN LENGTH OF LARGE-SCALE LLMS

| Model | Method | Number of turns | | | | |
|---|---|---|---|---|---|---|
| | | **0** | **1** | **2** | **3** | **4** |
| Llama-3.1-70B | Multi-turn | 231.0 | 715.1 | 1373.4 | 2070.4 | 2902.1 |
| | Reflexion | 231.0 | 740.4 | 1059.7 | 1301.0 | 1340.6 |
| | In-place | 231.0 | 378.0 | 472.7 | 534.1 | 610.2 |
| Gemma-3-27B | Multi-turn | 177.9 | 1833.3 | 3527.8 | 4934.1 | 6250.7 |
| | Reflexion | 177.9 | 1897.3 | 1769.4 | 1958.0 | 2020.5 |
| | In-place | 177.9 | 1008.8 | 1425.0 | 1540.9 | 1731.2 |

Table A16: Input token length on the Math-Hard dataset.

| Model | Method | Number of turns | | | | |
|---|---|---|---|---|---|---|
| | | **0** | **1** | **2** | **3** | **4** |
| Llama-3.1-70B | Multi-turn | 165.1 | 747.6 | 1342.3 | 1987.7 | 2608.3 |
| | Reflexion | 165.1 | 776.8 | 954.3 | 1187.8 | 1210.4 |
| | In-place | 165.1 | 451.9 | 515.8 | 560.3 | 595.4 |
| Gemma-3-27B | Multi-turn | 149.0 | 894.6 | 1672.0 | 2454.1 | 3209.8 |
| | Reflexion | 149.0 | 924.6 | 1117.7 | 1315.8 | 1314.0 |
| | In-place | 149.0 | 473.6 | 553.6 | 642.9 | 649.5 |

Table A17: Input token length on the MMLU-Pro dataset.

| Model | Method | Number of turns | | | | |
|---|---|---|---|---|---|---|
| | | **0** | **1** | **2** | **3** | **4** |
| Llama-3.1-70B | Multi-turn | 224.1 | 998.2 | 1720.3 | 2405.4 | 3099.9 |
| | Reflexion | 224.1 | 1027.5 | 1138.1 | 1309.9 | 1307.2 |
| | In-place | 224.1 | 585.9 | 636.0 | 694.4 | 752.2 |
| Gemma-3-27B | Multi-turn | 205.0 | 1085.9 | 1903.6 | 2754.2 | 3606.0 |
| | Reflexion | 205.0 | 1115.9 | 1230.8 | 1380.2 | 1414.2 |
| | In-place | 205.0 | 555.0 | 692.9 | 756.2 | 783.1 |

Table A18: Input token length on the GPQA dataset.

# D QUALITATIVE EXAMPLES

## D.1 IN-PLACE FEEDBACK EXAMPLE

Figure A9, Figure A10, Figure A11, and Figure A12 are the qualitative examples of in-place feedback on the three benchmarks.

## D.2 MULTI-TURN FEEDBACK FAILURE EXAMPLE

We observe failure cases of multi-turn feedback, and present the instances in Figure A13, Figure A14, and Figure A15.

---

**LLM-as-a-judge prompt for MATH-hard**

**SYSTEM PROMPT**
You are a mathematical equivalence judge.
Given a question, a correct answer, and a model's prediction, determine if they are mathematically equivalent even if they have different formatting or representation.
Respond with ONLY 'YES' if they are equivalent, or 'NO' if they are not.
If the model's prediction is correct, but the formatting is wrong, please respond with 'YES'.
DO NOT RESPOND WITH ANYTHING ELSE.

- - - - - - - - - - - - - - - - - - - - - - - - - - - - - - - - - - - -

**USER PROMPT**
Question: {QUESTION}
Correct answer: {CORRECT_ANSWER}
Model prediction: {PREDICTION}
Are these equivalent? Answer YES or NO.

(a) Prompt format used in MATH-hard.

---

**LLM-as-a-judge prompt for MMLU-pro free-form and GPQA free-form**

**SYSTEM PROMPT**
You are a strict and impartial judge for evaluating model predictions.
Given a question, a correct answer, and a model's prediction, decide whether the prediction is equivalent to the correct answer.
Respond with ONLY 'YES' or 'NO'. Never output anything else.

**Judgment Criteria**
1. The prediction must fully answer the question and cannot be vague.
2. Ignore differences in formatting, punctuation, capitalization, or spacing.
3. For numeric answers:
- Compute the relative error = |prediction − ground truth| / mean(prediction, ground truth).
- The prediction is correct if the relative error < 1%.
- If the ground truth is a single numeric value but the prediction is a range, treat it as incorrect (even if the range contains the ground truth).
- If the ground truth is a single value but the prediction is a range, output 'NO'.
4. If the prediction is correct in substance but expressed in a different format, respond 'YES'.

Final Rule: Output must be exactly 'YES' or 'NO'. No explanations, no extra words.

- - - - - - - - - - - - - - - - - - - - - - - - - - - - - - - - - - - -

**USER PROMPT**
Question: {QUESTION}
Correct answer: {CORRECT_ANSWER}
Model prediction: {PREDICTION}
Are these equivalent? Answer YES or NO.

(b) Prompt format used in MMLU-pro free-form and GPQA free-form.

Figure A3: Prompt format used for LLM-as-a-judge in MATH-hard, MMLU-pro free-form, and GPQA free-form.

# E   THE USE OF LARGE LANGUAGE MODELS

We used an LLM assistant as a writing tool for grammar checking and paraphrasing. In addition, since our experiments required carefully designed prompts, we employed the assistant to refine prompts.

**Prompt format used for feedback agent**

**SYSTEM PROMPT**

You are a rigorous verifier of LLM answers.
Your task: Given a question, the model's full answer, and the correct answer, identify the first and most critical reasoning flaw that causes the error.
You MUST NOT reveal or speculate about the correct final answer.
- Note that the options in previous questions might have been switched in each different attempt.

Your output must follow the structure below:
Explain in (maximum 2–3) precise sentences why that step is incorrect. Be specific in your critique.
Give short and specific actionable advice to fix it.

Constraints:
You may only give feedback on one reasoning error at a time — the earliest critical one.
If the model's answer is correct in logic but wrong in formatting, your feedback should address the formatting issue.
You DO NOT leak and mention the correct final answer.
**DO NOT SOLVE THE PROBLEM, JUST PROVIDE FEEDBACK WITHOUT MENTIONING THE CORRECT ANSWER**
Your output strictly starting with "Your output is wrong because" and then followed by the explanation.

Do not output anything else.

- - - - - - - - - - - - - - - - - - - - - - - - - - - - - - - - - - - - - - - - - - - - - - - - - - -

**USER PROMPT**

Your goal is to identify the **earliest and most critical logical flaw** in the model's reasoning
DO NOT provide or hint at the correct final answer.
- Note that the options in previous questions might have been switched in each different attempt.

**Problem**
{PROBLEM}

**Most Recent Model Answer**
{MODEL ANSWER}

**Correct Final Answer**
{GROUND TRUTH}

Based on this, provide feedback on the single most important error in the model's answer.
Do not leak and mention the correct final answer and do not add any extra commentary.

Figure A4: Prompt format used for feedback agent in MATH-hard, MMLU-pro free-form, and GPQA free-form.

---

**Prompt format used for in-place feedback agent**

**SYSTEM PROMPT**

You are a helpful assistant who INTERVENES in a math solution based on user feedback.

Your job is to produce a JSON object for a single **in-place replace** operation:

- Identify the **shortest unique substring (SUS)** from the original solution that must be edited to apply the feedback.
- Produce the revised text for that exact span.
- **Do NOT** change anything before the flaw, and **do NOT** continue solving the problem beyond where the feedback applies.
- **Preserve** all whitespace, punctuation, LaTeX, and casing exactly as in the original solution for the target substring.
- The target must be a **contiguous** substring that occurs **exactly once** in the original solution. If not unique, minimally extend the span (e.g., include adjacent tokens or punctuation) until it becomes unique.
- Return **ONLY** valid JSON with UTF-8 and proper escaping (no trailing commas, no extra commentary).

Return JSON with this schema (single edit only):

```
{
    "target_sentence": "< exact shortest unique substring copied from the original>",
    "edit_sentence": "<the revised substring after applying the feedback>"
}
```

Constraints:
- Output must be a single-line or multi-line JSON object; do not include any extra text.
- Do not normalize quotes/hyphens/spaces; copy exactly from the original for target_sentence.
- Do not introduce additional edits beyond the specified span.
- Do not provide a reasoning process beyond the feedback.

- - - - - - - - - - - - - - - - - - - - - - - - - - - - - - - - - - - - - - - - - - - - - - -

**USER PROMPT**

<The Start of Answer>
{ANSWER}
<The End of Answer>
<The Start of Original Solution>
{ORIGINAL_SOLUTION}
<The End of Original Solution>
<The Start of User Feedback>
{USER_FEEDBACK}
<The End of User Feedback>
<The Start of Instructions>
Write the JSON according to the following:
- Apply ONLY the given feedback to the original solution.
- Identify the **shortest unique substring** in the original that must change to satisfy the feedback; this must appear **exactly once**.
- If the obvious sentence occurs multiple times, **minimally extend** the span (e.g., prepend/append one or two nearby tokens or punctuation) until uniqueness holds.
- Put the original substring in "target_sentence" (copied **verbatim** from the original, including whitespace/newlines).
- Put the corrected version in "edit_sentence".
- If the feedback is sentence-like, keep it within **three sentences** in the edited span.
- STRICTLY FOLLOW:
- **Do not solve the problem** beyond where the feedback applies.
- Stop right after applying the feedback.
- Return **ONLY** valid JSON with keys "target_sentence" and "edit_sentence".

Figure A5: Prompt format used for in-place feedback agent in MATH-hard, MMLU-pro free-form, and GPQA free-form.

---

### Prompt format used for ZebraLogic

**SYSTEM PROMPT**
You are a helpful assistant that solves zebra puzzles.
Given a puzzle and a json template, you need to solve the puzzle and fill in the json template.
You need to fill in the json template with the correct attributes.

- - - - - - - - - - - - - - - - - - - - - - - - - - - - - - - - - - - - - - - - - - -

**USER PROMPT**
# Example Puzzle

There are 3 houses, numbered 1 to 3 from left to right, as seen from across the street. Each house is occupied by a different person.
Each house has a unique attribute for each of the following characteristics:
- Each person has a unique name: 'Peter', 'Eric', 'Arnold'.
- Each person has a unique favorite drink: 'tea', 'water', 'milk'

## Clues for the Example Puzzle

1. Peter is in the second house.
2. Arnold is directly left of the one who only drinks water.
3. The one who only drinks water is directly left of the person who likes milk.

## Answer to the Example Puzzle

```
{{
    "solution": {{
        "House 1": {{
            "Name": "Arnold",
            "Drink": "tea"
        }},
        "House 2": {{
            "Name": "Peter",
            "Drink": "water"
        }},
        "House 3": {{
            "Name": "Eric",
            "Drink": "milk"
        }}
    }}
}}
```

# Puzzle to Solve

{PUZZLE}

# Instruction

Now please solve the above puzzle. Present your reasoning and solution in the following json format:
{JSON TEMPLATE}

Figure A6: Input template used for ZebraLogic.

> **Feedback template for `Gemma` and `Qwen` in ZebraLogic**
>
> Your answer is incorrect. Please revise your solution based on the following feedback.
> - '{CATEGORY}' of the '{HOUSE}' is '{GROUND TRUTH}', not '{PREDICTION}'.
> - ...

(a) Feedback template for `Gemma` and `Qwen`.

> **Feedback template for `Llama` in ZebraLogic**
>
> Please revise your step-by-step reasoning based on the following feedback, and then provide a solution in the following json format.
> Do not just provide the final solution, but also provide the reasoning process.
>
> - '{CATEGORY}' of the '{HOUSE}' is '{GROUND TRUTH}', not '{PREDICTION}'.
> - ...

(b) Feedback template for `Llama`.

Figure A7: Rule based feedback template in ZebraLogic.

---

**Prompt format used for in-place feedback agent in ZebraLogic**

**SYSTEM PROMPT**

You are an in-place patcher. Given a reasoning step and feedback items, decide if the step conflicts with the feedback. If so, produce the minimally edited step that preserves all non-conflicting content and enforces the feedback exactly.

LOCKED CLUE SPANS
- Any substring beginning with 'Clue' and ending at the first period is read-only. Do not alter or delete it.
- If a step contains locked span(s), edit only after the last locked span.
- If formatted as "Clue N: ... . - editable text", edit only after the first ' - '. If editable text is deleted, remove the dangling ' - '.

REFERENCE ATTRIBUTES
{CATEGORY}
- Use this only as a reference to detect conflicts and apply equivalence normalization. Do not generate new information beyond feedback.

EDIT RULES
- Feedback is always given in positive-only form, e.g., 'attr' of the 'house i' is 'A'.
- Always overwrite the attribute with the provided value A.
- Modify only attributes mentioned in the feedback.
- Do not add new reasoning, deductions, constraints, or explanations.
- Do not insert additional sentences beyond the minimal correction.
- Do not add new information, negations, or style fixes.
- Equivalence normalization:
* 'name' of house i is X $\leftrightarrow$ X is in House
* 'musicgenre' of house i is Y, The Y house is House i
* Apply the same for other attributes.
- Rewrite conflicts to the positive corrected assignment. Never use negations.

CONFLICT CRITERIA
A statement conflicts only if it:
- Mentions the same (house, attribute) with a contradictory value, or
- Assigns a feedback value to the wrong house by equivalence.

DEPENDENCY PRUNING
- If a sentence depends on a premise that is corrected or negated by the feedback and cannot be minimally rewritten, delete that sentence only.
- Do not delete the entire reasoning step.
- Never alter or delete locked clue text.

OUTPUT
- If no conflict: "conflict":"no"
- If conflict: "conflict":"yes","replacement":"..."
where replacement is the fully patched step, including unchanged locked clues.

RULES
- Make the smallest edits necessary.
- Preserve capitalization, spacing, and order.
- Never output empty replacement if a locked clue exists.
- Absolutely never add new reasoning, deductions, or commentary not present in the original.

- - - - - - - - - - - - - - - - - - - - - - - - - - - - - - - - - - - - - - - - -

**USER PROMPT**
Reasoning step
{REASONING_STEP}
Feedback
{FEEDBACK}

---

Figure A8: Prompt format used for in-place feedback agent in ZebraLogic.

---

**In-place feedback example - MATH-hard**

**Problem**
Express the infinite series $\frac{3}{206} + \frac{9}{2\cdot103^2} + \frac{27}{2\cdot103^3} + \cdots$ as a terminating decimal.

- - - - - - - - - - - - - - - - - - - - - - - - - - - - - - - - - - - - - - - -

**Model response**
Let $S = \frac{3}{206} + \frac{9}{2\cdot103^2} + \frac{27}{2\cdot103^3} + \cdots$. We can rewrite this as

$$S = \sum_{n=1}^{\infty} \frac{3^n}{2 \cdot 103^{n+1}} = \frac{1}{2} \sum_{n=1}^{\infty} \frac{3^n}{103^{n+1}} = \frac{1}{2} \sum_{n=1}^{\infty} \frac{3^n}{103 \cdot 103^n} = \frac{1}{2 \cdot 103} \sum_{n=1}^{\infty} \left(\frac{3}{103}\right)^n$$

The sum $\sum_{n=1}^{\infty} \left(\frac{3}{103}\right)^n$ is a geometric series with first term $\frac{3}{103}$ and common ratio $\frac{3}{103}$. Since $\left|\frac{3}{103}\right| <$ 1, the series converges to

$$\sum_{n=1}^{\infty} \left(\frac{3}{103}\right)^n = \frac{\frac{3}{103}}{1 - \frac{3}{103}} = \frac{\frac{3}{103}}{\frac{103-3}{103}} = \frac{\frac{3}{103}}{\frac{100}{103}} = \frac{3}{100}$$

Therefore, $S = \frac{1}{2 \cdot 103} \cdot \frac{3}{100} = \frac{3}{20600}$ Now we need to express $\frac{3}{20600}$ as a terminating decimal. We can divide 3 by 20600: $\frac{3}{20600} = 0.00014645\ldots$ We can write $\frac{3}{20600} = \frac{3}{206\cdot100} = \frac{3}{206} \cdot \frac{1}{100}$. We have $\frac{3}{206} \approx 0.01464539$. Then $\frac{3}{20600} = \frac{3}{206} \cdot \frac{1}{100} \approx 0.01464539 \cdot \frac{1}{100} = 0.0001464539$. We can also write $\frac{3}{20600} = \frac{3}{206\cdot100} = \frac{3}{206} \cdot \frac{1}{100}$. We can divide 3 by 206 to get 0.01464539.... Then we multiply by $\frac{1}{100}$ to get 0.0001464539.... We can write $\frac{3}{20600} = 0.0001464539...$ The decimal representation is 0.0001464539.... We want to express this as a terminating decimal. We have $S = \frac{3}{20600}$. Dividing 3 by 20600, we get $3 \div 20600 = 0.0001464539...$ Since we want a terminating decimal, we need to find a common denominator. We have $20600 = 2^3 \cdot 5^2 \cdot 103$. So $\frac{3}{20600} = \frac{3}{2^3\cdot5^2\cdot103}$. We can write $\frac{3}{20600} = 0.0001464539...$ The decimal representation is 0.0001464539.... Final Answer: The final answer is $\boxed{0.000146}$

- - - - - - - - - - - - - - - - - - - - - - - - - - - - - - - - - - - - - - - -

**1) In-place edit**
Let $S = \frac{3}{206} + \frac{9}{2\cdot103^2} + \frac{27}{2\cdot103^3} + \cdots$. We can rewrite this as $S = \sum_{n=1}^{\infty} \frac{3^n}{2\cdot103^n} = \frac{1}{2} \sum_{n=1}^{\infty} \left(\frac{3}{103}\right)^n$

- - - - - - - - - - - - - - - - - - - - - - - - - - - - - - - - - - - - - - - -

**2) Continuation generation**
This is a geometric series with first term $a = \frac{3}{103}$ and common ratio $r = \frac{3}{103}$. Since $|r| = \frac{3}{103} < 1$, the series converges and its sum is given by

$$\sum_{n=1}^{\infty} \left(\frac{3}{103}\right)^n = \frac{\frac{3}{103}}{1 - \frac{3}{103}} = \frac{\frac{3}{103}}{\frac{103-3}{103}} = \frac{\frac{3}{103}}{\frac{100}{103}} = \frac{3}{100}$$

Therefore, $S = \frac{1}{2} \cdot \frac{3}{100} = \frac{3}{200}$ To express this as a terminating decimal, we divide 3 by 200: $\frac{3}{200} = \frac{3}{2\cdot100} = \frac{3}{2\cdot10^2} = \frac{3}{2} \cdot \frac{1}{100} = 1.5 \cdot 0.01 = 0.015$ Alternatively, we can directly divide 3 by 200: $\frac{3}{200} = 0.015$ Thus, the infinite series is equal to $\boxed{0.015}$

Figure A9: In-place feedback example for the MATH-hard dataset.

---

**In-place feedback example - MATH-hard**

**Problem**

Dr. Zaius invests $10,000$ in a CD with an annual interest rate of $4\%$ that compounds semi-annually (twice a year). After six months, he rolls over the CD into another CD with an annual interest rate of $5\%$ that also compounds semi-annually. After six months in the second CD, how much does Dr. Zaius have, in dollars?

- - - - - - - - - - - - - - - - - - - - - - - - - - - - - - - - - - - - - - - - - - - - - - - - - - - - -

**Model response**

First, we need to calculate the amount after the first six months with the first CD. The principal is $10000$, the annual interest rate is $4\%$, and the interest compounds semi-annually. So the interest rate per period is $4\%/2 = 2\% = 0.02$. The number of periods is $6/2 = 3$. The formula for compound interest is $A = P(1+r)^n$, where $A$ is the amount after $n$ periods, $P$ is the principal, and $r$ is the interest rate per period. In this case, $P = 10000$, $r = 0.02$, and $n = 3$. So, $A = 10000(1+0.02)^3 = 10000(1.02)^3 = 10000(1.061208) = 10612.08$. After six months, Dr. Zaius rolls over the CD into another CD with an annual interest rate of $5\%$ that also compounds semi-annually. The principal is $10612.08$. The annual interest rate is $5\%$, so the interest rate per period is $5\%/2 = 2.5\% = 0.025$. The number of periods is $6/2 = 3$. Using the same formula, $A = P(1+r)^n$, we have $P = 10612.08$, $r = 0.025$, and $n = 3$. So, $A = 10612.08(1 + 0.025)^3 = 10612.08(1.025)^3 = 10612.08(1.076890625) \approx 11418.53$.

Therefore, after six months in the second CD, Dr. Zaius has approximately $\boxed{11418.53}$

- - - - - - - - - - - - - - - - - - - - - - - - - - - - - - - - - - - - - - - - - - - - - - - - - - - - -

**1) In-place edit**

First, we need to calculate the amount after the first six months with the first CD. The principal is $10000$, the annual interest rate is $4\%$, and the interest compounds semi-annually. So the interest rate per period is $4\%/2 = 2\% = 0.02$. The number of periods is $1$.

- - - - - - - - - - - - - - - - - - - - - - - - - - - - - - - - - - - - - - - - - - - - - - - - - - - - -

**2) Continuation generation**

The amount after six months is $P(1 + r)^n = 10000(1 + 0.02)^1 = 10000(1.02) = 10200$. Then, Dr. Zaius rolls over the CD into another CD with an annual interest rate of $5\%$ that also compounds semi-annually. The principal is $10200$, the annual interest rate is $5\%$, and the interest rate per period is $5\%/2 = 2.5\% = 0.025$. The number of periods is $1$. The amount after six months in the second CD is $10200(1 + 0.025)^1 = 10200(1.025) = 10200 + 10200 \times 0.025 = 10200 + 255 = 10455$.

So, after six months in the second CD, Dr. Zaius has $\boxed{10455}$

Figure A10: In-place feedback example for the MATH-hard dataset.

1512
1513
1514
1515
1516
1517
1518
1519
1520
1521
1522
1523
1524
1525
1526
1527
1528
1529
1530
1531
1532
1533
1534
1535
1536
1537
1538
1539
1540
1541
1542
1543
1544
1545
1546
1547
1548
1549
1550
1551
1552
1553
1554
1555
1556
1557
1558
1559
1560
1561
1562
1563
1564
1565

### In-place feedback example - GPQA free-form

**Problem**

Consider an isolated system of 13 identical particles each with charge 2e and negligible mass. 12 of these charges are constrained to stay at 2 m from a point P. The 13th charge is fixed at P. What is the minimum energy (in Joules) of this system correct to three decimals?

- - - - - - - - - - - - - - - - - - - - - - - - - - - - - - - - - - - - - - - - - - - - - - - - - -

**Model response**

Let $q$ be the charge of each particle, and $r$ be the distance of the 12 particles from the point P. The 13th charge is at the point P. The Coulomb force between two charges $q_1$ and $q_2$ separated by a distance $r$ is given by $F = \frac{kq_1q_2}{r^2}$, where $k = \frac{1}{4\pi\epsilon_0}$ is the Coulomb constant. *[omitted for brevity...]*

The potential energy of the system is given by

$$U = \sum_{i=1}^{12} \frac{kq^2}{r_i}$$

where $r_i$ is the distance from the $i$-th charge to the point P. Since all 12 charges are at a distance of 2 m from P, $r_i = 2$ for all $i$. $U = 12 \cdot \frac{kq^2}{2} = 6kq^2$ *[omitted for brevity...]*

The total energy is $U = 6kq^2$. $U = 6 \cdot \frac{1}{4\pi\epsilon_0}(2e)^2 = \frac{6\cdot4e^2}{4\pi\epsilon_0} = \frac{6e^2}{\pi\epsilon_0} \approx \frac{6(1.602\times10^{-19})^2}{\pi(8.854\times10^{-12})} \approx \frac{6(2.5664\times10^{-38})}{\pi(8.854\times10^{-12})} \approx \frac{15.40\times10^{-38}}{2.783\times10^{-11}} \approx 5.57 \times 10^{-27}J$. The minimum energy is approximately

$5.570e - 27J$

- - - - - - - - - - - - - - - - - - - - - - - - - - - - - - - - - - - - - - - - - - - - - - - - - -

**1) In-place edit**

Let $q$ be the charge of each particle, and $r$ be the distance of the 12 particles from the point P. The 13th charge is at the point P. The Coulomb force between two charges $q_1$ and $q_2$ separated by a distance $r$ is given by $F = \frac{kq_1q_2}{r^2}$, where $k = \frac{1}{4\pi\epsilon_0}$ is the Coulomb constant. *[omitted for brevity...]*

The potential energy of the system is given by

$$U = \sum_{i<j} \frac{kq_iq_j}{r_{ij}}$$

- - - - - - - - - - - - - - - - - - - - - - - - - - - - - - - - - - - - - - - - - - - - - - - - - -

**2) Continuation generation**

which includes the 12 central–outer terms and the 66 outer–outer pairwise terms; the outer–outer distances $r_{ij}$ depend on the arrangement of the 12 charges on the sphere of radius 2, so one must evaluate this pairwise sum for the energy-minimizing configuration before computing any numeric value. *[omitted for brevity...]*

The minimum energy is $U = \frac{13}{2}\frac{kq^2}{r} = \frac{13}{2}\frac{\frac{1}{4\pi\epsilon_0}(2e)^2}{2} = \frac{13e^2}{4\pi\epsilon_0} \approx \frac{13(1.602\times10^{-19})^2}{4\pi(8.854\times10^{-12})} \approx 2.82 \times 10^{-26}J$.

$2.820e - 26J$

Figure A11: In-place feedback example for the GPQA free-form dataset.



**In-place feedback example - MMLU-pro free-form**

**Problem**
A particle of mass $m$ and velocity $u_1$ makes a head-on collision with another particle of mass $2m$ at rest. If the coefficient of restitution is such to make the loss of total kinetic energy a maximum, what are the velocities $v_1$ after the collision?

---

**Model response**
Let $m_1 = m$ and $v_1 = u_1$ be the mass and initial velocity of the first particle. Let $m_2 = 2m$ and $v_2 = 0$ be the mass and initial velocity of the second particle. Let $v_1'$ and $v_2'$ be the velocities of the first and second particles after the collision, respectively. The initial total kinetic energy is $K_i = \frac{1}{2}m_1 u_1^2 + \frac{1}{2}m_2(0)^2 = \frac{1}{2}mu_1^2$. The final total kinetic energy is $K_f = \frac{1}{2}m_1 v_1' + \frac{1}{2}m_2 v_2'$. The loss of kinetic energy is $\Delta K = K_i - K_f = \frac{1}{2}mu_1^2 - \frac{1}{2}mv_1' - \frac{1}{2}(2m)v_2'$. *[omitted for brevity...]*

To maximize $\Delta K$, we need to minimize $v_2'$. Since $v_2' = \frac{1}{2}v_1'$, we have $v_2' = \frac{1}{2}v_1'$. Then $v_1' = 2v_2'$. $\Delta K = \frac{1}{2}mu_1^2 - 2m\frac{1}{2}v_1' = \frac{1}{2}mu_1^2 - mv_1'$. Final Answer: The final answer is $\boxed{0}$

---

**1) In-place edit**
Let $m_1 = m$ and $v_1 = u_1$ be the mass and initial velocity of the first particle. Let $m_2 = 2m$ and $v_2 = 0$ be the mass and initial velocity of the second particle. Let $v_1'$ and $v_2'$ be the velocities of the first and second particles after the collision, respectively. The initial total kinetic energy is $K_i = \frac{1}{2}m_1 u_1^2 + \frac{1}{2}m_2(0)^2 = \frac{1}{2}mu_1^2$. The final total kinetic energy is $K_f = \frac{1}{2}m_1 v_1'^2 + \frac{1}{2}m_2 v_2'^2$.

---

**2) Continuation generation**
The loss of kinetic energy is $\Delta K = K_i - K_f = \frac{1}{2}mu_1^2 - \frac{1}{2}m_1 v_1'^2 - \frac{1}{2}m_2 v_2'^2$. We want to maximize $\Delta K$. The coefficient of restitution is $e = \frac{v_2' - v_1'}{u_1 - v_2'}$. Since it's a head-on collision, we can use the conservation of momentum: $m_1 u_1 + m_2(0) = m_1 v_1' + m_2 v_2'$ $mu_1 = mv_1' + 2mv_2'$ $u_1 = v_1' + 2v_2'$ $v_2' = \frac{u_1 - v_1'}{2}$ *[omitted for brevity...]*

To maximize $\Delta K$, we take the derivative with respect to $v_1'$ and set it to zero. However, we are given that the loss of kinetic energy is maximized. This occurs when $\Delta K$ is maximized. For maximum loss of kinetic energy, we want $e$ to be as small as possible, i.e., $e = 0$. If $e = 0$, then $v_1' = \frac{u_1(1-0)}{0+3} = \frac{u_1}{3}$.

Then $v_2' = \frac{u_1 - v_1'}{2} = \frac{u_1 - \frac{u_1}{3}}{2} = \frac{\frac{2u_1}{3}}{2} = \frac{u_1}{3}$. $\boxed{v_1' = \frac{u_1}{3}}$



Figure A12: In-place feedback example for the MMLU-pro free-form dataset.



**Failure example of multi-turn refinement - 1) corrupting correct content**

**Problem**

In convex hexagon $ABCDEF$, all six sides are congruent, $\angle A$ and $\angle D$ are right angles, and $\angle B$, $\angle C$, $\angle E$, and $\angle F$ are congruent. The area of the hexagonal region is $2116(\sqrt{2}+1)$. Find $AB$.

- - - - - - - - - - - - - - - - - - - - - - - - - - - - - - - - - - - - - - - - - -

**Model response**

To solve for $AB$ in the given convex hexagon $ABCDEF$, we will follow these steps:

1. **Identify the properties of the hexagon:**
- All six sides are congruent.
- $\angle A$ and $\angle D$ are right angles.
- $\angle B$, $\angle C$, $\angle E$, and $\angle F$ are congruent.

2. **Determine the measure of the congruent angles:**
- Since the sum of the interior angles of a hexagon is $(6-2) \times 180° = 720°$.
- Let $\angle B = \angle C = \angle E = \angle F = x$.
- We have $\angle A = 90°$ and $\angle D = 90°$.
- Therefore, $4x + 180° = 720°$.
- Solving for $x$: $4x = 540° \implies x = 135°$

3. **Visualize the hexagon:** - Place the hexagon on a coordinate plane with $A$ at $(0,0)$ and $D$ at $(a,0)$.
- Since $\angle A = 90°$ and $\angle D = 90°$, the hexagon can be divided into a rectangle and two right triangles.

4. **Divide the hexagon into simpler shapes:**
- The hexagon can be divided into a rectangle $ABCD$ and two right triangles $ABE$ and $CDF$.

Since the problem asks for $AB$, and we have determined that $AB = s$, the final answer is: $\boxed{23}$

- - - - - - - - - - - - - - - - - - - - - - - - - - - - - - - - - - - - - - - - - -

**Feedback**

Your output is wrong because you assumed without justification that the hexagon **"can be divided into a rectangle and two right triangles" and then treated** $ABCD$ **as a rectangle (even as a square) with both width and height equal to** $s$**; right angles at** $A$ **and** $D$ **alone do not imply** $AB \parallel CD$ **or** $BC \parallel AD$**, so** $ABCD$ **need not be a rectangle and the triangle legs need not equal** $s$**.**

Fix it by producing a correct partition justified by the given angles (for example, draw and justify specific diagonals such as $BF$ and $CE$), use the 135° measure of the other four angles to deduce the shapes and side relationships of the pieces, and compute the area from that valid decomposition rather than assuming a rectangle/square.

- - - - - - - - - - - - - - - - - - - - - - - - - - - - - - - - - - - - - - - - - -

**Model response**

Let's re-evaluate the problem step-by-step, ensuring we correctly decompose the hexagon based on the given angles and properties.

1. **Identify the properties of the hexagon:**
- All six sides are congruent.
- $\angle A = 90°$ and $\angle D = 90°$.
- $\angle B = \angle C = \angle E = \angle F = 135°$.

2. **Visualize the hexagon:**
- Place the hexagon on a coordinate plane with $A$ at $(0,0)$ and $D$ at $(s,0)$.

3. **Draw and justify specific diagonals:**
- Draw diagonal $BF$ and $CE$. These diagonals will help us decompose the hexagon into simpler shapes.

4. **Analyze the resulting shapes:**
- Diagonal $BF$ splits $\angle B$ and $\angle F$ into two angles each: $\angle ABF = \angle FBD = 45°$.
- Similarly, diagonal $CE$ splits $\angle C$ and $\angle E$ into two angles each: $\angle BCE = \angle ECD = 45°$.

Since the problem asks for $AB$, and we have determined that $AB = s$, the final answer is: $\boxed{23}$



Figure A13: Failure example of multi-turn refinement on the MATH-hard free-form dataset: corrupting correct content.

---

### Failure example of multi-turn refinement - 2) ignoring feedback

**Problem**

A container weighs 3.22 lb force when empty. Filled with water at 60°F the mass of the container and its contents is 1.95 slugs. Find the weight of the water in the container and its volume in cubic feet. Assume density of water= 62.4 lb force/ft$^3$.

- - - - - - - - - - - - - - - - - - - - - - - - - - - - - - - - - - - - - - - - - - - - - - - - - - -

**Model response**

Let $W_c$ be the weight of the empty container in lb force, which is given as $W_c = 3.22$ lb. Let $m$ be the mass of the container and its contents in slugs, which is given as $m = 1.95$ slugs. Let $\rho$ be the density of water in lb force/ft$^3$, which is given as $\rho = 62.4$ lb/ft$^3$. We want to find the weight of the water in the container, $W_w$, and the volume of the water in the container, $V$ in cubic feet.

First, we find the weight of the container and its contents due to gravity. We know that weight $W = mg$, where $m$ is mass and $g$ is the acceleration due to gravity. We can use the given mass in slugs and the standard value of $g = 32.2$ ft/s$^2$. $W = m \cdot g = 1.95$ slugs $\cdot 32.2$ ft/s$^2 = 62.19$ lb.

The weight of the container and its contents is also the sum of the weight of the empty container and the weight of the water. $W_c + W_w = 62.19$ lb $3.22 + W_w = 62.19$ $W_w = 62.19 - 3.22 = 58.97$ lb.

Now we can find the volume of the water in the container using the density formula: $\rho = \frac{W}{V}$, where $W$ is the weight of the water and $V$ is the volume. $V = \frac{W}{\rho} = \frac{58.97 \text{ lb}}{62.4 \text{ lb/ft}^3} \approx 0.9455$ ft$^3$.

Therefore, the weight of the water is 58.97 lb and the volume of the water is approximately 0.9455 ft$^3$.

> Weight of water $= 58.97$ lb, Volume $= 0.9455$ ft$^3$

- - - - - - - - - - - - - - - - - - - - - - - - - - - - - - - - - - - - - - - - - - - - - - - - - - -

**Feedback**

**Your output is wrong because you miscomputed the total weight: you evaluated $W = m \cdot g$ incorrectly (you used 1.95·32.2 = 62.19), which led to an incorrect water weight. Recompute $W = m \cdot g$ with the correct product, then subtract the empty container weight to get the water weight and divide that by the given density to obtain the volume.**

- - - - - - - - - - - - - - - - - - - - - - - - - - - - - - - - - - - - - - - - - - - - - - - - - - -

**Model response**

Let $W_c$ be the weight of the empty container in lb force, which is given as $W_c = 3.22$ lb. Let $m$ be the mass of the container and its contents in slugs, which is given as $m = 1.95$ slugs. Let $\rho$ be the density of water in lb force/ft$^3$, which is given as $\rho = 62.4$ lb/ft$^3$. We want to find the weight of the water in the container, $W_w$, and the volume of the water in the container, $V$ in cubic feet.

First, we find the weight of the container and its contents due to gravity. We know that weight $W = mg$, where $m$ is mass and $g$ is the acceleration due to gravity. We can use the given mass in slugs and the standard value of $g = 32.2$ ft/s$^2$. $W = m \cdot g = 1.95$ slugs $\cdot 32.2$ ft/s$^2 = 62.19$ lb.

The weight of the container and its contents is 62.19 lb. The weight of the empty container is 3.22 lb. The weight of the water is the difference between the total weight and the weight of the empty container: $W_w = 62.19 - 3.22 = 58.97$ lb.

Now we can find the volume of the water in the container using the density formula: $\rho = \frac{W}{V}$, where $W$ is the weight of the water and $V$ is the volume. $V = \frac{W}{\rho} = \frac{58.97 \text{ lb}}{62.4 \text{ lb/ft}^3} \approx 0.9455$ ft$^3$.

Therefore, the weight of the water is 58.97 lb and the volume of the water is approximately 0.9455 ft$^3$.

> Weight of water $= 58.97$ lb, Volume $= 0.9455$ ft$^3$

Figure A14: Failure example of multi-turn refinement on the MMLU-pro free-form dataset: ignoring feedback.

---

### Failure example of multi-turn refinement - 3) introducing a new error

**Problem**

A 125/250-volt three-wire system has a load on the positive side of 500 amp, on the negative side of 450 amp, the neutral current therefore being 50 amp. If each machine of the balancer set has an efficiency of 86 percent, calculate the current for each of these machines and for the main generator.

- - - - - - - - - - - - - - - - - - - - - - - - - - - - - - - - - - - - - - - - - - -

**Model response**

Here's a step-by-step solution to the problem:

1. **Calculate the total power in the positive phase:**
$P_{pos} = V_{pos} \times I_{pos} = 125V \times 500A = 62500W$
2. **Calculate the total power in the negative phase:**
$P_{neg} = V_{neg} \times I_{neg} = 125V \times 450A = 56250W$
3. **Calculate the total power in the neutral phase:**
$P_{neutral} = V_{neutral} \times I_{neutral} = 125V \times 50A = 6250W$
4. **Calculate the total system power:**
$P_{total} = P_{pos} + P_{neg} + P_{neutral} = 62500W + 56250W + 6250W = 125000W$
5. **Determine the number of balancer sets:**
The problem states each balancer set has an efficiency of 86%, which implies that there are two balancer sets. *[omitted for brevity...]*

The balancer sets are designed to correct the imbalance. The generator provides 1000A. The imbalance is 500A - 450A = 50A. The balancer sets must provide 50A. Since each set has 86% efficiency, the current each set provides is $I = \frac{50}{0.86} \approx 581.39A$. So, $I_1 = I_2 = 581.39A$.

$I_{generator} = 1000, I_{set1} = 581.39, I_{set2} = 581.39$

- - - - - - - - - - - - - - - - - - - - - - - - - - - - - - - - - - - - - - - - - - -

**Feedback**

Your output is wrong because **you infer there are two balancer sets from the 86% efficiency — efficiency does not determine the number of machines, so splitting total power by two is an unjustified non sequitur that corrupts all subsequent calculations.** To fix it, do not assume a quantity: either use the actual number of machines stated in the problem or introduce a variable n, first compute the required balancing current from the load imbalance (or neutral current), then divide that required output among n machines and account for each machine's 86% efficiency by dividing the required output current/power by 0.86 to obtain the input/current for each machine.

- - - - - - - - - - - - - - - - - - - - - - - - - - - - - - - - - - - - - - - - - - -

**Model response**

Okay, you are absolutely correct. My apologies for the flawed reasoning. Let's redo the calculation with the correct approach.

1. **Calculate the total power**:
$P_{total} = 62500W + 56250W + 6250W = 125000W$
2. **Calculate the imbalance:**
The imbalance is the difference between the positive and negative currents: $500A - 450A = 50A$. This is the current the balancer sets need to correct.
3. **Determine the number of balancer sets:**
The problem doesn't explicitly state the number of balancer sets. Let's denote the number of balancer sets as $n$.
4. **Calculate the required current per balancer set:**
Each balancer set needs to provide $50A$.
5. **Calculate the current required from each balancer set, accounting for efficiency:**
Current per set = (Required current) / (Efficiency) = $50A/0.86 = 581.395A$ (approximately)

$I_{generator} = 1000, I_{set1} = 58.14, I_{set2} = 58.14$

---

Figure A15: Failure example of multi-turn refinement on the MMLU-pro free-form dataset: introducing a new error.

