# OpenReview forum: "In-Place Feedback: A New Paradigm for Guiding Large Language Models in Multi-Turn Reasoning"
_ICLR.cc/2026/Conference — ICLR 2026 Conference Withdrawn Submission_

### Official Review · Reviewer_pZrn · 2025-10-16

**Soundness:** 3
**Presentation:** 2
**Contribution:** 2
**Rating:** 4
**Confidence:** 3

**Summary:**

This work introduces in-place feedback, an interaction method where users directly edit a large language model’s (LLM) previous response, and the model generates a new output conditioned on the edited context. This approach achieves stronger refinement performance on reasoning benchmarks while being more token-efficient—requiring fewer input and output tokens.

Overall, I believe the paper’s main weakness lies in the lack of compelling real-world applications, and I have listed several concerns below. At this stage, I do not think the paper meets the acceptance threshold; however, if the authors can provide strong and convincing responses to these issues, I would be willing to raise my score.

**Strengths:**

1. The paper is clearly written and easy to understand.
2. The proposed method is novel and may provide some insights for the community.
3. The experiments are substantial.

**Weaknesses:**

1. The practical value of this paper seems limited. In many cases, users ask LLMs questions about topics they themselves don’t fully understand, making it difficult to provide effective feedback—or even to determine whether the model’s response is correct or incorrect. Consequently, the proposed method appears to lack compelling real-world applications.
2. In-place feedback essentially provides the model with more prior information, so it's not surprising that it outperforms standard multi-turn interaction. In contrast, some approaches—such as [1]—improve response quality without requiring external human input, which arguably offers greater practical value.

[1] Madaan, Aman, et al. "Self-refine: Iterative refinement with self-feedback." Advances in Neural Information Processing Systems 36 (2023): 46534-46594.

3. Regarding the experimental results, readers would likely appreciate seeing precise numerical values; therefore, presenting the findings exclusively through figures may not be the most effective approach.

**Questions:**

1. The paper’s evaluation appears to be conducted primarily on tasks with known ground-truth answers. However, it remains unclear how the proposed method would perform on datasets where the correct answer is not available—such as in real-world, partially observable, or proprietary settings. Addressing this gap would be important for assessing the method’s applicability in more practical, open-ended scenarios.

2. In practice, users most commonly rely on larger-sized models. Has the author experimented with the proposed method on very large models (e.g., those with tens or hundreds of billions of parameters)?

3. How exactly is the pruning of reasoning context relevant to the edited segment, as mentioned in Section 2.3, performed?

---

> ### Author Response · Authors · 2025-11-20
>
> We thank you for sharing your insightful and constructive feedback. We address your concerns in the following responses.
>
> ---
>
> > The practical value of this paper seems limited. In many cases, users ask LLMs questions about topics they themselves don’t fully understand, making it difficult to provide effective feedback—or even to determine whether the model’s response is correct or incorrect. Consequently, the proposed method appears to lack compelling real-world applications.
>
> > The paper’s evaluation appears to be conducted primarily on tasks with known ground-truth answers. However, it remains unclear how the proposed method would perform on datasets where the correct answer is not available—such as in real-world, partially observable, or proprietary settings. Addressing this gap would be important for assessing the method’s applicability in more practical, open-ended scenarios.
>
> We kindly ask the reviewer to refer to our general comment. In short, we consider an expert-in-the-loop setting, where users can reliably identify specific errors and provide localized feedback. In such scenarios, the model’s ability to faithfully incorporate this feedback is essential for preserving the productivity gains and task acceleration.
>
> Regarding the experimental settings, we respectfully note that our experiments focus on settings where both answers and feedback are objectively verifiable, and we acknowledge that this makes the scope somewhat narrow. Our intention was to evaluate the core mechanism under conditions where correctness can be clearly determined. Extending in-place feedback to broader, open-ended reasoning tasks is substantially more difficult because feedback quality cannot be guaranteed and evaluation becomes inherently ambiguous. As a future direction, we are considering examining more complex scenarios through targeted user studies with domain experts.
>
>
> > In-place feedback essentially provides the model with more prior information, so it's not surprising that it outperforms standard multi-turn interaction. In contrast, some approaches—such as [1]—improve response quality without requiring external human input, which arguably offers greater practical value.
> [1] Madaan, Aman, et al. "Self-refine: Iterative refinement with self-feedback." Advances in Neural Information Processing Systems 36 (2023): 46534-46594.
>
> We respectfully clarify that our research scope is explicitly centered on how to incorporate external feedback to improve productivity in expert-in-the-loop scenarios, rather than on eliminating the need for such feedback. Prior work has shown that relying solely on self-generated feedback is fundamentally limited: current LLMs often fail to reliably self-correct their own reasoning errors without external signals [1]. These findings highlight the necessity of external feedback even in the presence of self-refinement mechanisms. Furthermore, external feedback remains valuable even when models attempt self-refinement, as human or expert-provided edits can correct the very errors that self-feedback mechanisms fail to detect.
>
>
> 1. Large Language Models Cannot Self-Correct Reasoning Yet, ICLR 2024
>
>
> > Regarding the experimental results, readers would likely appreciate seeing precise numerical values; therefore, presenting the findings exclusively through figures may not be the most effective approach.
>
> We appreciate the reviewer’s suggestion. In the revised version, we will include comprehensive tables reporting the exact scores for all settings, in addition to the existing figures, so that readers can clearly see the numerical improvements.
>
> > In practice, users most commonly rely on larger-sized models. Has the author experimented with the proposed method on very large models (e.g., those with tens or hundreds of billions of parameters)?
>
>
> We further expand our evaluation to larger models such as Llama-3.1-70B-Instruct and Gemma-3-27B-Instruct. Due to computational and cost constraints, we evaluate these large-scale models for five turns. The combined tables integrating the new experiments with the previously reported results are provided in Appendix.C of the revised paper. We find that in-place feedback consistently continues to outperform multi-turn prompting.

---

> > ### Author Response · Authors · 2025-11-20
> >
> > > How exactly is the pruning of reasoning context relevant to the edited segment, as mentioned in Section 2.3, performed?
> >
> > Since each benchmark has different reasoning-structure characteristics, our pruning procedure adapts to the task accordingly. For MATH-hard, MMLU-pro, and GPQA, where the reasoning is highly sequential, the in-place agent prunes by truncating the entire suffix following the edited span. Since the in-place agent provides localized feedback on a specific erroneous segment, all subsequent steps must be regenerated from that corrected point to maintain consistency.
> >
> > For ZebraLogic, the reasoning is partially parallel since many reasoning steps can operate on different cells and do not depend on each other. To handle this structure, the in-place agent first segments the output into per-step reasoning blocks and compares each block against the feedback. It revises only the steps that conflict with the corrected values and prunes downstream steps whose correctness depends on those values, while leaving independent steps intact. After applying dependency-based pruning, the in-place agent also removes the final JSON answer block to ensure both reasoning and structured outputs are regenerated from the corrected state.

---

> > > ### Comment · Reviewer_pZrn · 2025-11-27
> > > **Reply to the rebuttal**
> > >
> > > Thanks for the author' reply.
> > >
> > > I noticed that almost everyone pointed out that the methods were not practical enough.  In practice, people usually choose to use more powerful closed-source models, such as the latest GPT5 and Gemini3, rather than the few small open-source models used in the authors' experiments. Furthermore, the authors used closed-source models to replace human editors in the paper. Does this mean that the authors also believe these closed-source models are close to or even surpass human capabilities? If so, the authors' method seems to lose its meaning.

---

> > > > ### Author Response · Authors · 2025-11-27
> > > >
> > > > Thank you for your response.
> > > >
> > > > Let us restate the goal of our work. We propose In-Place Feedback as an effective way for users to communicate with an LLM when they have enough knowledge to understand the model’s response. Regardless of the base LLM’s size, if its answer contains incorrect information and the user can detect it, we believe In-Place Feedback can operate more effectively than conventional multi-turn prompting. Although we could not test this with GPT-5 due to the lack of continuation-generation api, we show that In-Place Feedback remains effective even for larger-scale models, and we expect the same to hold for closed-source models like GPT-5. Please refer to Appendix C.2 and our general comment for more details.
> > > >
> > > > Additionally, as discussed in the general comment, our experiments assume users can sufficiently understand the model’s output. We used GPT-5 as a proxy for users only because running existing benchmarks with real users would be prohibitively costly. We believe that, with real human participants, In-Place Feedback would achieve similar or even better performance.

---

### Official Review · Reviewer_6ugG · 2025-10-28

**Soundness:** 4
**Presentation:** 4
**Contribution:** 3
**Rating:** 6
**Confidence:** 4

**Summary:**

This paper investigates in-place feedback as an alternative feedback method to conversational, multi-turn feedback. Empirically, the paper shows in-place feedback is more effective than multi-turn feedback, and has additional benefits in saving compute. Section 4 presents an in-depth analysis on ZebraLogic, using 5 metrics to diagnose how models incorporate feedback, and further validated the benefits of in-place feedback.

**Strengths:**

This is a well-written and well-executed paper. The paper takes an unique perspective and studies the interface for language models to incorporate feedback. I'm convinced that in-place feedback is a useful and effective method, and I find many experiments and findings in the paper interesting and insightful.

**Weaknesses:**

* Currently the paper uses gpt-5-mini to automate the human users who provide feedback. This is a reasonable approximation but also creates a gap towards the claimed use case. The paper could be further improved by doing small batch experiments with real humans providing feedback to LLMs.
* Additionally the datasets used in the paper (MMLU, MATH, GPQA) are very different from real-world use cases where human inspects LLM answers and provide feedback. Validation in more realistic settings could be helpful or worth investigating in future work.

**Questions:**

* How does gpt-5-mini perform on the datasets you used? Since it's used to provide feedback to other models, it will be helpful to report its performance and consider it as an upper bound.
* Does in-place feedback work in a "self-refine" setting? Currently you used gpt-5-mini as the "teacher" and three weaker models as the "student". I wonder if this method is still more useful when the teacher and the student are the same model.
* Do you have any hypothesis on _why_ in-place feedback is more effective than multi-turn feedback? E.g., Is this relevant to any known limitations or mechanisms of transformers?

(Willing to adjust my rating if these questions are addressed.)

---

> ### Author Response · Authors · 2025-11-20
>
> We sincerely appreciate your thoughtful comments, and we have provided detailed replies to each concern below.
>
> ---
>
> > How does gpt-5-mini perform on the datasets you used? Since it's used to provide feedback to other models, it will be helpful to report its performance and consider it as an upper bound.
>
> We clarify that gpt-5-mini receives the problem, the model’s response, and the ground-truth answer when generating feedback. Prior work has shown that this setup yields nearly perfect feedback [1], so we expect its performance to be close to 100%. Nevertheless, if the reviewer finds it helpful, we can measure performance of gpt-5-mini and report it.
>
> 1. FEEDBACK FRICTION: LLMs Struggle to Fully Incorporate External Feedback, NeurIPS 2025
>
> > Does in-place feedback work in a "self-refine" setting? Currently you used gpt-5-mini as the "teacher" and three weaker models as the "student". I wonder if this method is still more useful when the teacher and the student are the same model.
>
> We appreciate suggesting valuable direction. We note that closed-source LLMs do not support text continuation, which prevents us from directly applying in-place feedback in a self-refinement setting. We are conducting experiments with a large-scale open-source model to evaluate in-place self-correction, and if these experiments complete within the rebuttal period, we will upload the results immediately.
>
>
> > Do you have any hypothesis on why in-place feedback is more effective than multi-turn feedback? E.g., Is this relevant to any known limitations or mechanisms of transformers?
>
>
> A plausible hypothesis is that LLMs employ induction head mechanisms that reinforce and extend previously generated patterns [1]. When applied to multi-turn refinement, such mechanisms tend to propagate earlier mistakes forward, which coherently explains the empirical pattern observed in Figure 6: CTRR and FAR decreases across turns as the model repeatedly regenerates revised response conditioned on earlier and wrong responses.
>
> In contrast, in-place feedback does not accumulate additional context. It regenerates only within the edited span, which prevents the model from repeatedly conditioning on earlier deviations and amplifying them. This lack of context accumulation is consistent with the higher CTRR and FAR. Understanding these behaviors through the lens of induction-head mechanisms provides a plausible explanation for why in-place feedback is more stable than multi-turn refinement.
>
> 1. In-context Learning and Induction Heads, Arxiv 2022
>
>
> > Currently the paper uses gpt-5-mini to automate the human users who provide feedback. This is a reasonable approximation but also creates a gap towards the claimed use case. The paper could be further improved by doing small batch experiments with real humans providing feedback to LLMs.
>
> > Additionally the datasets used in the paper (MMLU, MATH, GPQA) are very different from real-world use cases where human inspects LLM answers and provide feedback. Validation in more realistic settings could be helpful or worth investigating in future work.
>
> We agree with the reviewer’s point. As mentioned in other comments as well, running human studies on a wider range of tasks would help reduce the gap between simulated and real user feedback. However, conducting such a study during the rebuttal period is not feasible given the time constraints. We are considering adding such studies in our follow-up revision.

---

> > ### Comment · Reviewer_6ugG · 2025-11-28
> >
> > Thank you for your detailed response. I've also read the general response. I appreciate the new experiments using larger models and including Reflexion for comparison. I also appreciate the discussion on attributing the effectiveness to induction heads of language models.
> >
> > I agree with the authors that "expert-in-the-loop" represents a realistic and important setting for human-AI collaboration. In my opinion, using AI coding tools like Cursor and write code collaboratively is a form in-place feedback, and investigating interaction methods for these scenarios is meaningful.
> >
> > I do note that the experimental setup presents two limitations: (1) using language models to simulate human experts, and (2) evaluating on tasks below expert level. This creates a sim-to-real gap. To me, this is acceptable since the authors are doing an early exploration of an interaction method. The paper is self-contained and it's reasonable to further scale things up in a follow-up study.

---

> > > ### Author Response · Authors · 2025-11-28
> > >
> > > Thank you for the supportive and thoughtful comments. We appreciate your perspective on the importance of studying “expert-in-the-loop” scenarios, and we are glad that you view our work as a meaningful early exploration for human–AI collaboration.
> > >
> > > We agree with your observation regarding the sim-to-real gap. In the revision, we will address this by incorporating an initial human study to help narrow the gap. Separately, we will also include the self-refine experiments you suggested.
> > >
> > > Once again, thank you for the constructive feedback. Your comments meaningfully guide our planned revision.

---

### Official Review · Reviewer_Xh9G · 2025-10-30

**Soundness:** 2
**Presentation:** 2
**Contribution:** 2
**Rating:** 2
**Confidence:** 4

**Summary:**

This paper investigates an improved interaction paradigm for large language models (LLMs) in multi-turn reasoning settings, in-place feedback. This method enables users directly edit the model’s previous response, and the LLM generates a revision based on this modified version. Through experiments on multiple reasoning-intensive benchmarks, the paper shows that in-place feedback yields better performance and higher feedback precision compared to conventional multi-turn feedback, while also reducing token usage. Additional analyses in controlled environments demonstrate that this approach effectively mitigates a key weakness of standard feedback — the model’s tendency to misapply corrections or introduce new errors. Overall, the study presents in-place feedback as a more natural and efficient strategy for guiding LLMs in complex reasoning tasks.

**Strengths:**

1. This paper presents a very simple method, which makes it easy to follow.
2. The case figures in this paper are very clear and effectively illustrate the main ideas presented in the work.
3. The experimental setup is straightforward and reproducible.

**Weaknesses:**

1. The proposed approach demonstrates limited innovation and seems to focus more on an engineering-oriented implementation than on advancing fundamental research.Describing it as a new paradigm may be overstated.
2. In Figure 1, the third failure mode — “the feedback is applied but causes errors in subsequent reasoning steps” — does not appear to be unique to multi-turn feedback. Such errors are often uncontrollable, as large language models themselves tend to produce mistakes progressively even when the earlier reasoning steps are correct. Therefore, categorizing this issue as a failure mode specific to multi-turn feedback does not seem reasonable.
3. The proposed in-place edit approach’s interaction mode shifts much of the difficulty in problem-solving to the user. In a standard multi-turn setting, users only need to provide feedback on what the model misunderstood (as shown in Figure 1, where the model misinterprets “2 more balls are added to the total” as “each”). In contrast, the proposed method requires users to fully understand each step of the model’s reasoning, identify the specific errors, and then modify them directly. This paradigm may work for relatively simple mathematical problems where users can easily spot mistakes, but it becomes impractical for more complex problems in which users themselves may not be able to detect the model’s errors.

**Questions:**

N/A

---

> ### Author Response · Authors · 2025-11-20
>
> We appreciate your valuable feedback and have addressed all points of concern in our responses.
>
> ---
>
> > The proposed in-place edit approach’s interaction mode shifts much of the difficulty in problem-solving to the user. In a standard multi-turn setting, users only need to provide feedback on what the model misunderstood (as shown in Figure 1, where the model misinterprets “2 more balls are added to the total” as “each”). In contrast, the proposed method requires users to fully understand each step of the model’s reasoning, identify the specific errors, and then modify them directly. This paradigm may work for relatively simple mathematical problems where users can easily spot mistakes, but it becomes impractical for more complex problems in which users themselves may not be able to detect the model’s errors.
>
> We first kindly ask the reviewer to refer to our general comment. In brief, our work explicitly targets an expert-in-the-loop setting in which users can reliably identify specific errors and provide localized feedback. This assumption is shared with standard multi-turn feedback methods, which also presuppose that users can recognize erroneous span. Our contribution is to make such feedback more direct and targeted by allowing edits on the exact erroneous span.
>
> Regarding the concern about human labor, prior HCI work shows that UI-level systems supporting localized, in-place edits already help users work more efficiently and with lower cognitive load, since they can directly pinpoint and correct errors [1,2,3]. Although these benefits arise from interface design, they demonstrate that localized editing is a practical and effective workflow in real applications. Our work aligns with these observations by exploring how similar advantages can be realized at the model level rather than through a new interface.
>
>
> 1. DirectGPT: A Direct Manipulation Interface to Interact with Large Language Models, CHI 2024
> 2. Interactive Reasoning: Visualizing and Controlling Chain-of-Thought Reasoning in Large Language Models, Arxiv 2025
> 3. WaitGPT: Monitoring and Steering Conversational LLM Agent in Data Analysis with On-the-Fly Code Visualization, UIST 2024
>
>
>
> > The proposed approach demonstrates limited innovation and seems to focus more on an engineering-oriented implementation than on advancing fundamental research. Describing it as a new paradigm may be overstated.
>
> We clarify that our contribution focuses on introducing a conceptual shift in how feedback is provided to LLMs, rather than proposing an entirely new algorithmic pipeline. While most existing approaches rely on prompt-based feedback, our work explores edit-based feedback, a fundamentally different way of grounding user guidance directly in the model’s response. The term “new paradigm” is used to highlight this shift in interaction design, not to overstate the contribution. To better reflect this intent, we are considering revising the title accordingly, such as “In-Place Feedback: Directly Editing LLM Responses for Guiding LLMs in Multi-Turn Reasoning.”
>
>
> > In Figure 1, the third failure mode — “the feedback is applied but causes errors in subsequent reasoning steps” — does not appear to be unique to multi-turn feedback. Such errors are often uncontrollable, as large language models themselves tend to produce mistakes progressively even when the earlier reasoning steps are correct. Therefore, categorizing this issue as a failure mode specific to multi-turn feedback does not seem reasonable.
>
> We agree that the third failure mode is not unique to multi-turn feedback. However, our analysis shows that this phenomenon is *amplified* in multi-turn settings, as reflected in the **Correction Through Reasoning Ratio (CTRR)**. CTRR measures how effectively a model generalizes the given feedback to correct *non-targeted reasoning errors* in subsequent steps. As shown in Figure 6, models become increasingly prone to generating new downstream errors under multi-turn feedback, which in turn leads to a consistent decline in CTRR across turns. In contrast, in-place feedback sustains higher CTRR values by constraining regeneration to the corrected state, which limits unintended drift in later reasoning. Thus, while such cascading errors can occur in any feedback mode, our results suggest they are substantially *worsened* in multi-turn refinement, precisely the issue captured by CTRR.

---

### Official Review · Reviewer_x7Uc · 2025-10-30

**Soundness:** 2
**Presentation:** 2
**Contribution:** 1
**Rating:** 2
**Confidence:** 3

**Summary:**

Researchers found that letting users directly edit an AI's wrong answers works much better than just telling the AI "that's wrong, fix it" - the AI fixes mistakes more accurately, uses 79% fewer tokens, and doesn't accidentally break parts that were already correct.

**Strengths:**

1. **Thorough testing** - Multiple datasets and models show consistent improvements
2. **Identifies why failures happen** - Pinpoints exactly how traditional feedback breaks (corrupting correct parts, ignoring feedback, creating new errors)

**Weaknesses:**

### 1. Critical methodology gap with "user edits"
The paper states multiple times that the method involves "user directly edit an LLM's previous response" and "user feedback is applied as an edit," but in the real experiments, the authors used GPT-5-mini instead as the human editor.
If the authors intended to use real human/users as the editors, the method requires significant human labor to identify error locations and provide precise edits, which limits real-world applicability. If the authors intended to use an AI agent as an alternative to the human editors, then the whole writing in the paper has a huge flaw, where throughout the paper, it shows the advantages of involving real users in the feedback editing stage.
There is no additional training for GPT-5-mini to identify which parts in the response need to be replaced. Meanwhile, GPT-5-mini is rather a strong model compared to the models in the paper (Gemma-3-4b, Qwen2.5-7B, Llama-3.1-8B). An ablation study is needed to show if different models used in the editing stage would change the final result. If the in-place feedback models are the weaker models like Gemma-3-4b, Qwen2.5-7B, Llama-3.1-8B, would it still bring improvement to the tasks?
### 2. Missing key related work
The paper is missing key related work, such as the intrinsic self-correction idea in https://arxiv.org/abs/2310.01798. At the same time, while the proposed method highlighted the advantages of letting users directly edit the LLM's responses, recent works (https://arxiv.org/abs/2409.12822) demonstrate that LMs can produce hard-to-detect errors, which raises questions about the reliability of human-provided in-place edits. The authors should make more discussion on how human-provided in-place edits would work in the mentioned tasks and on the failure modes of human-provided in-place edits.
### 3. Flawed token efficiency analysis
The token efficiency part in Line 243 makes faulty analysis. It says "For input tokens, multi-turn feedback appends new turns to the dialogue history, causing token usage to grow linearly with the number of turns" and "multi-turn feedback accumulates the full dialogue history." In early literature in 2023, there are already papers on multi-turn/iterative reasoning that don't keep the whole dialogue history as the prompt, for example Reflexion (https://arxiv.org/abs/2303.11366), MemGPT (https://arxiv.org/abs/2310.08560), MemPrompt (https://aclanthology.org/2022.emnlp-main.183.pdf). The paper also lacks proper citations.
### 4. Limited experimental scope and presentation
No analysis of how the method scales to longer, more complex tasks. Task selection is narrow, focusing primarily on mathematical and logical reasoning without exploring other domains. The chosen datasets do not cover variations of reasoning scenarios and can hardly be called comprehensive experiments. Testing is restricted to relatively small open-source models (Gemma-3-4b, Qwen2.5-7B, Llama-3.1-8B). There is no evaluation on state-of-the-art models, more recent reasoning-optimized models, or proprietary models like GPT and Claude. Additionally, the paper doesn't have a comprehensive table for all results, making it hard to see what their improvements are by percentage.

**Questions:**

1. This paper identifies failure modes as corrupting correct parts, ignoring feedback, and creating new errors. Have you counted what percentage of the 3 different failure modes occur in different tasks, and to what extent do the baseline and your approach fix these failure modes?
2. Recent literature found that multi-turn refinement usually stops improving after the first round. What is the first-round performance for your in-place feedback approach?
3. What are the reasons if, after one round of in-place feedback, the answer is still incorrect? What failure modes remain?
4. Have you tested scenarios where the feedback itself is incorrect or misleading? How robust is in-place feedback to erroneous edits?
5. Have you analyzed the quality of edits made by GPT-5-mini? How often does it correctly identify the minimal edit span versus over-editing or under-editing?

---

> ### Author Response · Authors · 2025-11-20
>
> We sincerely appreciate your thoughtful comments, and we have provided detailed replies to each concern below.
>
> ---
>
> > **Critical methodology gap with "user edits"**
> > The paper states multiple times that the method involves "user directly edit an LLM's previous response" and "user feedback is applied as an edit," but in the real experiments, the authors used GPT-5-mini instead as the human editor. If the authors intended to use real human/users as the editors, the method requires significant human labor to identify error locations and provide precise edits, which limits real-world applicability. If the authors intended to use an AI agent as an alternative to the human editors, then the whole writing in the paper has a huge flaw, where throughout the paper, it shows the advantages of involving real users in the feedback editing stage.
>
> We emphasize that our method is designed for human editors rather than AI agents, and we kindly ask the reviewer to refer to our general comment. In brief, we consider an expert-in-the-loop setting in which users can reliably identify specific errors and provide localized feedback. In such scenarios, the model’s ability to faithfully incorporate external feedback is essential to preserving the intended productivity gains and task acceleration.
>
> Regarding the concern about human labor, prior HCI work has shown that UI-level systems supporting localized, in-place edits help users complete tasks more efficiently and with lower cognitive load, as users can directly pinpoint and correct errors [1,2,3]. Although these benefits come from interface design rather than model-level mechanisms, they demonstrate that localized editing is a practical and effective workflow in real applications. Our work aligns with these observations by exploring how similar advantages can be realized at the model level rather than through a new interface. We also point out that user feedback carries a similar amount of information as in traditional multi-turn interactions.
>
>
> 1. DirectGPT: A Direct Manipulation Interface to Interact with Large Language Models, CHI 2024
> 2. Interactive Reasoning: Visualizing and Controlling Chain-of-Thought Reasoning in Large Language Models, Arxiv 2025
> 3. WaitGPT: Monitoring and Steering Conversational LLM Agent in Data Analysis with On-the-Fly Code Visualization, UIST 2024
>
> > There is no additional training for GPT-5-mini to identify which parts in the response need to be replaced. Meanwhile, GPT-5-mini is rather a strong model compared to the models in the paper (Gemma-3-4b, Qwen2.5-7B, Llama-3.1-8B). An ablation study is needed to show if different models used in the editing stage would change the final result. If the in-place feedback models are the weaker models like Gemma-3-4b, Qwen2.5-7B, Llama-3.1-8B, would it still bring improvement to the tasks?
>
> We would like to clarify that *GPT-5-mini is positioned as the expert simulator* in this framework, responsible for providing feedback within a practical scenario. Our observations indicate that low-capacity LLMs, such as Gemma-3-4b, Qwen2.5-7B, and Llama-3.1-8B, perform suboptimally in the in-place editing task. Therefore, using low-capacity models as an expert simulator is not a suitable approach to reflect our target scenario.

---

> ### Author Response · Authors · 2025-11-20
>
> > **Limited experimental scope and presentation**  & **Flawed token efficiency analysis**
> No analysis of how the method scales to longer, more complex tasks. Task selection is narrow, focusing primarily on mathematical and logical reasoning without exploring other domains. The chosen datasets do not cover variations of reasoning scenarios and can hardly be called comprehensive experiments. Testing is restricted to relatively small open-source models (Gemma-3-4b, Qwen2.5-7B, Llama-3.1-8B). There is no evaluation on state-of-the-art models, more recent reasoning-optimized models, or proprietary models like GPT and Claude. Additionally, the paper doesn't have a comprehensive table for all results, making it hard to see what their improvements are by percentage.
>
> > The token efficiency part in Line 243 makes faulty analysis. It says "For input tokens, multi-turn feedback appends new turns to the dialogue history, causing token usage to grow linearly with the number of turns" and "multi-turn feedback accumulates the full dialogue history." In early literature in 2023, there are already papers on multi-turn/iterative reasoning that don't keep the whole dialogue history as the prompt, for example Reflexion, MemGPT, MemPrompt. The paper also lacks proper citations.
>
>
>
> We kindly remind that our experiments focus on settings where both answers and feedback are objectively verifiable, and we acknowledge that this makes the scope somewhat narrow. Our intention was to evaluate the core mechanism under conditions where correctness can be clearly determined. Extending in-place feedback to broader, open-ended reasoning tasks is substantially more difficult because feedback quality cannot be guaranteed and evaluation becomes inherently ambiguous. As a future direction, we are considering examining more complex scenarios through targeted user studies with domain experts.
>
> In addition, we appreciate the reviewer for pointing out the missing related work on iterative refinement methods that do not maintain the full dialogue history. We will add appropriate references to multi-turn refinement methods that do not accumulate the full dialogue history, such as Reflexion, MemGPT, MemPrompt, in the revised manuscript. In response to this comment, we additionally include Reflexion as a memory-based refinement baseline in our experiments. We further expand our evaluation to larger models such as Llama-3.1-70B-Instruct and Gemma-3-27B-Instruct. We report the combined tables of the new experiments and the previously reported results in Appendix.C of the revised paper.
>
> For Reflexion, we preserve up to three feedbacks. Due to computational and cost constraints, we evaluate the large-scale models for five turns. We find that in-place feedback continues to outperform both multi-turn prompting and Reflexion in these settings. It also provides higher token efficiency even when the context is not accumulated, which suggests that the benefit of in-place editing is not limited to scenarios where multi-turn baselines retain the entire conversation.
>
> > **Missing key related work.** The paper is missing key related work, such as the intrinsic self-correction idea in https://arxiv.org/abs/2310.01798. At the same time, while the proposed method highlighted the advantages of letting users directly edit the LLM's responses, recent works (https://arxiv.org/abs/2409.12822) demonstrate that LMs can produce hard-to-detect errors, which raises questions about the reliability of human-provided in-place edits. The authors should make more discussion on how human-provided in-place edits would work in the mentioned tasks and on the failure modes of human-provided in-place edits.
>
> We appreciate the reviewer’s insightful comment and have incorporated the suggested references into the revised manuscript. We agree that recent studies highlight the issue of subtle, hard-to-detect errors in LLM outputs. However, this limitation is not specific to in-place editing. It is a fundamental challenge shared by all human-in-the-loop feedback paradigms, including multi-turn prompting. Our contribution is to enhance feedback efficiency by anchoring user edits to the exact target erroneous span. This localized edit mechanism makes user feedback more effective.
>
> Additionally, we recognize that incorrectly provided edits could propagate incorrect corrections. While we do not directly address this issue in the current work, we discuss potential mitigation strategies, such as adding a verification or self-checking stage, in the section on ethical considerations. We view this as an important direction for future research.

---

> > ### Author Response · Authors · 2025-11-20
> >
> > > This paper identifies failure modes as corrupting correct parts, ignoring feedback, and creating new errors. Have you counted what percentage of the 3 different failure modes occur in different tasks, and to what extent do the baseline and your approach fix these failure modes?
> >
> > The three failure modes we describe are empirical patterns we repeatedly observed across tasks, but quantifying them reliably is difficult. For this reason, a core part of our study is to design an evaluation setting that can systematically analyze these failure behaviors. Specifically, we instead adopt ZebraLogic as an automated and reproducible way to evaluate whether feedback is incorporated correctly.
> >
> > Figure 6 provides evidence on why LLMs fail to incorporate feedback. The three failure patterns persist even as turns accumulate in standard multi-turn refinement. In contrast, in-place feedback exhibits two key advantages: 1) higher feedback acceptance ratio in later turns, and 2) more effective continuation of reasoning after a feedback edit compared to standard multi-turn refinement. We kindly refer the reviewer to Section 4.2 for a more detailed discussion.
> >
> > > Recent literature found that multi-turn refinement usually stops improving after the first round. What is the first-round performance for your in-place feedback approach?
> >
> > We report the task performance in the Figure 3 or the table in Appendix C.1. We note that, compared with multi-turn refinement, our approach achieves faster performance gains in the rounds, reaching higher accuracy with fewer interaction steps.
> >
> > > What are the reasons if, after one round of in-place feedback, the answer is still incorrect? What failure modes remain?
> >
> > Even after one round of in-place feedback, the model can still produce an incorrect final answer if subsequent reasoning steps introduce new failures, for example, when later parts of the chain of thought deviate from the corrected segment. In other words, in-place feedback does not completely eliminate the possibility of new errors arising in the following reasoning steps. However, as shown in Figure 6, In-place feedback substantially improves the reliability of these later reasoning steps compared to the multi-turn refinement, leading to fewer such failures.
> >
> > > Have you tested scenarios where the feedback itself is incorrect or misleading? How robust is in-place feedback to erroneous edits?
> >
> > No, we have not. We kindly remind that our target research scenario is an expert-in-the-loop setting, where the expert is assumed to provide high-quality and reliable feedback. For this reason, incorrect or misleading feedback is outside the scope of our target problem. However, we acknowledge that erroneous edits could propagate additional mistakes. Although we do not address this case empirically in the current work, we expect that incorporating a verification or self-checking stage could serve as a promising mitigation strategy in settings where the reliability of feedback is uncertain.
> >
> >
> > > Have you analyzed the quality of edits made by GPT-5-mini? How often does it correctly identify the minimal edit span versus over-editing or under-editing?
> >
> > We have not conducted a dedicated qualitative study evaluating how often GPT-5-mini identifies the minimal edit span versus over-editing or under-editing. However, our method is explicitly designed to mitigate such issues: the feedback processing stage constrains the editable region and guides the model to focus on local changes, thereby reducing the likelihood of both over-editing and under-editing in practice. For more details on this process, we kindly refer the reviewer to Appendix A.2.

---

### Author Response · Authors · 2025-11-20
**General Comment**

## Clarification of target scenario and motivation

We first sincerely appreciate the reviewers for their insightful and constructive feedback. We fully understand that reviewers may naturally have concerns about the practicality of our method, and we acknowledge that this concern stems from our fault. We recognize that the introduction could have offered clearer motivation and clarification on this aspect. We appreciate the opportunity to address this more concretely here.

We would like to clarify that, in real-world settings, users often rely on LLMs not since they lack the ability to do the work themselves, but since the model helps them carry out their work more efficiently. This is common in software engineering, legal drafting, and clinical documentation [1,2,3,4,5,6,7], where experts rely on LLMs to accelerate routine or time-consuming portions of their work. In these scenarios, even a small incorrect intermediate statement, such as a wrong assumption, a misapplied rule, or an inconsistent description, can affect all subsequent parts of the output. For this reason, users inevitably need to review the model’s output and directly modify or give feedback on the parts they recognize as incorrect.

Prior work consistently describes the workflow described in our work: developers detect flawed conditions or assumptions in AI-generated code and modify or give feedback expecting the rest of the code to be updated accordingly [1,2,3]; lawyers identify clauses that unintentionally alter legal meaning and modify or give feedback so that related sections are revised as well [4,5]; clinicians correct dosage or contraindication errors and expect the remainder of the summary to adjust appropriately [6,7]. In all of these cases, the practical efficiency depends on *whether the model faithfully incorporates the user’s feedback across the rest of the output.* When the system fails to do so, users should re-check or rewrite large portions of the output, which reduces the efficiency advantages that originally motivated the model’s use.

Thus, in such 'expert-in-the-loop' settings closely matching real-world usage, the key requirement is not to remove the human from the verification loop, but to ensure that once the user modifies or gives feedback on an incorrect part, the model consistently reflects that correction throughout its output. This capability is essential for maintaining correctness, avoiding unnecessary repeated iterations, and preserving the practical efficiency gains expected from LLM assistance.

On the other hand, we can also consider scenarios in which verifying the correctness of a response is easy, while generating a correct answer from scratch is infeasible or extremely difficult. For instance, many NP problems are hard to solve but allow polynomial-time verification once a solution is given. In such settings, in-place feedback could also be effectively applied.

To reflect expert-in-the-loop settings, we adopt a controlled experimental setup in which the feedback is assumed to be nearly perfect [8]. We conduct our experiments in settings where both the answers and the feedback are objectively verifiable. Our intention is to evaluate the core mechanism under conditions where correctness can be clearly determined.



---
## Additional experimental results

We conduct additional experiments requested by the reviewers, including 1) evaluating the performance of our method with large-scale LLMs and 2) comparing in-place feedback against other baseline. The updated results are included in Appendix C (Lines 810–1066) of the revised submission. We respectfully request the reviewers to refer these new results. Please note that this is not the final revision. We are also revising the introduction and other sections highlighted in the reviews.

---
1. Let the Code LLM Edit Itself When You Edit the Code, ICLR 2025
2. EDIT-Bench: Evaluating LLMs on Instructed Code Edits, Arxiv 2025
3. TICODER: Test-Driven Interactive Code Generation, IEEE TSE 2024
4. Applying Large Language Models for Enhancing Contract Drafting, ICAIL Workshop LegalAIIA 2023
5. Applicability of LLMs for Legal Case Judgment Summarization, AI & Law 2024
6. EHR-Embedded LLM for Hospital-Course Summaries, JAMA Network Open 2025
7. LLM-Assisted Medical Documentation: Efficacy, Errors, & Ethics, Nature 2025
8. FEEDBACK FRICTION: LLMs Struggle to Fully Incorporate External Feedback, NeurIPS 2025

---

### Note · Authors · 2025-12-05

**Comment:**

After carefully considering the reviewers’ constructive feedback, we have decided to withdraw the current submission. The comments raised important points regarding the clarity of our problem setting and motivation, as well as the scope and completeness of our empirical evaluation.

We intend to revise the manuscript by clarifying the target scenario, underlying assumptions, and motivation of our work. In addition, we plan to further strengthen the experimental evaluation by incorporating larger-scale models, adding missing baselines, and considering additional datasets. These revisions will allow us to more thoroughly address the reviewers’ suggestions and to strengthen both the analysis and empirical evidence.

We sincerely appreciate the reviewers’ insightful feedback and plan to resubmit an improved version of this work in the future.

**Withdrawal Confirmation:**

I have read and agree with the venue's withdrawal policy on behalf of myself and my co-authors.